# Calibrating large-ensemble European climate projections using observational data

Christopher H. O'Reilly[1,2], Daniel J. Befort[1], and Antje Weisheimer[2,3]

[1]Atmospheric, Oceanic and Planetary Physics, Department of Physics, University of Oxford.
[2]NCAS-Climate, Department of Physics, University of Oxford.
[3]European Centre for Medium-Range Weather Forecasts (ECMWF).

**Correspondence:** Christopher H. O'Reilly (christopher.oreilly@physics.ox.ac.uk)

**Abstract.**

This study examines methods of calibrating projections of future regional climate for the next 40-50 years using large single model ensembles (the CESM Large Ensemble and MPI Grand Ensemble), applied over Europe. The three calibration methods tested here are more commonly used for initialised forecasts from weeks up to seasonal timescales. The calibration techniques are applied to ensemble climate projections, fitting seasonal ensemble data to observations over a reference period (1920-2016). The calibration methods were tested and verified using an "imperfect model" approach using the historical/RCP 8.5 simulations from the CMIP5 archive. All the calibration methods exhibit a similar performance, generally improving the out-of-sample projections in comparison to the uncalibrated (bias-corrected) ensemble. The calibration methods give results that are largely indistinguishable from one another, so the simplest of these methods, namely Homogeneous Gaussian Regression (HGR), is used for the subsequent analysis. As an extension to the HGR calibration method it is applied to dynamically decomposed data, in which the underlying data is separated into dynamical and residual components (HGR-decomp). Based on the verification results obtained using the imperfect model approach, the HGR-decomp method is found to produce more reliable and accurate projections than the uncalibrated ensemble for future climate over Europe. The calibrated projections for temperature demonstrate a particular improvement, whereas the projections for changes in precipitation generally remain fairly unreliable. When the two large ensembles are calibrated using observational data, the climate projections for Europe are far more consistent between the two ensembles, with both projecting a reduction in warming but a general increase in the uncertainty of the projected changes.

## 1 Introduction

To make informed assessments of climate impacts and implement relevant adaptation strategies, reliable climate projections are important for policy-makers and other stakeholders (e.g. Field et al., 2012). There is particular demand for climate projections on regional-scales for the next 40-50 years, however, such predictions are currently very uncertain (e.g. Stocker et al., 2013;

Knutti and Sedláček, 2013). One example of the demand for improved regional climate projections is the EU-funded "European Climate Prediction system" project (EUCP), which aims to produce reliable European climate projections from the present to the middle of the century (Hewitt and Lowe, 2018). In this study, which is a part of the EUCP project, we examine methods of improving the accuracy and reliability of climate projections over the European region.

There are a myriad of factors that contribute to the uncertainty in projections of future regional climate. One large factor is the uncertainty in greenhouse gas emissions and associated future radiative forcing anomalies (e.g. Pachauri et al., 2014). In this study we will focus only on estimating uncertainty of the physical climate system itself in responding to changing greenhouse gas forcing by focusing on a single representative concentration pathway (RCP), following the Coupled Model Intercomparison Project 5 (CMIP5) protocol (Taylor et al., 2012). The majority of analyses of coupled model projections are based upon multi-

model ensembles, which combine projections from multiple different coupled ocean-atmosphere climate models. One strength of a multi-model ensemble is that if each of the models has different structural deficiencies and associated errors, then these will not overly influence the ensemble projection. In seasonal forecasting, for example, multi-model products have been found to outperform the individual models in several studies (e.g. Palmer et al., 2004; Hagedorn et al., 2005; Baker et al., 2018). Multi-model ensembles of coupled climate models provide a range of plausible scenarios for the historical and future evolution

of the physical climate system. The simplest treatment of these models is to assume that each is equally likely, sometimes referred to as "model democracy" (e.g. Knutti, 2010). However, this approach assumes that models are independent and that they each represent an equally plausible representation of the climate system, neither of which is typically well justified (e.g. Gleckler et al., 2008; Knutti et al., 2013).

      Several methods have been developed that go beyond "model democracy" and instead weight models based on their per-

formance in an attempt to improve the representation of uncertainty in multi-model ensembles. One step away from model democracy is to downweight models in the ensemble that are not independent from one another (e.g. Sanderson et al., 2015), as has often found to be applicable in CMIP5-based studies. An additional or alternative approach is to weight models based on their past performance with respect to an observational benchmark, which could be the climatology of one or multiple fields (e.g. Giorgi and Mearns, 2002, 2003; Knutti et al., 2017; Sanderson et al., 2015; Merrifield et al., 2019) or the ability of models

to capture past changes (e.g. Kettleborough et al., 2007). In a recent paper, Brunner et al. (2019) applied a model weighting technique to the CMIP5 climate projections over the European region. The model weighting was found to constrain the large spread in the CMIP5 models and reduce the implied uncertainty in the multi-model projections of European climate over the coming decades.

      A weakness of multi-model ensembles, however, is that the different externally forced climate response in each of the models

can be difficult to isolate from internal variability. This is particularly problematic when each model typically only consists of a few ensemble members or less, as is the case with most models in CMIP5. To overcome the problem of disentangling the forced model response from the internal variability, several modelling groups have performed large single model ensemble simulations, using 40 ensemble members or more (e.g. Deser et al., 2020). When dealing with such large ensemble sizes, the ensemble mean provides a good estimate of the externally forced signal and deviations from this can reasonably be interpreted

as the internal variability of the coupled climate system. A further strength of large ensembles is that they can be used to

effectively attribute climate variability to changes in large-scale circulation. For example Deser et al. (2016) used a large ensemble to demonstrate that the observed wintertime temperature trends over the second half of the century were due to a combination of forced thermodynamic changes and a dynamically driven temperature trend that was not clearly externally forced. We will use the separation of the large ensembles into forced signals and internal variability, as well as the separation of each into dynamical and thermodynamical components, to examine different methods of calibrating projections of European climate.

Despite large ensembles providing clearer estimates of forced climate signal and internal variability, it is obvious that these ensembles will not perfectly represent observed climate variability, which is also the case with the multi-model ensembles. In this study, we explore the extent to which large ensemble climate projections can be calibrated over the observational period to adjust and potentially improve future projections. The general calibration approach relies upon the large ensemble being clearly separable into a forced signal component and residual internal variability (e.g. Deser et al., 2014). Calibration techniques have previously been applied to output from initialised seasonal forecasts (as well as shorter range forecasts), and have been demonstrated to reduce the forecast error and, perhaps more crucially, improve the reliability of the probabilistic forecasts (Kharin and Zwiers, 2003; Doblas-Reyes et al., 2005; Manzanas et al., 2019). In addition to seasonal timescales, calibration techniques have also been shown to be effective on the output from decadal prediction systems (Sansom et al., 2016; Pasternack et al., 2018). However, these types of ensemble calibration techniques have not previously been applied to ensemble climate model projections. Here we apply ensemble calibration techniques to uninitialised large ensemble climate projections, focusing on European regions, to test whether these ensembles can be calibrated to give reliable probabilistic climate projections for the next 40-50 years.

The paper is organised as follows. The datasets, verification techniques and calibration methods are described in the next section. In section 3, we present results from the different calibration methods, namely, "Variance Inflation", "Ensemble Model Output Statistics", and "Homogeneous Gaussian Regression". These calibration methods are applied to, and verified against, CMIP5 model data and also applied to observations. Conclusions follow in section 4.

## 2 Datasets and methods

### 2.1 Model and observational datasets

In this study we use two different large ensemble coupled climate model datasets. The first is from the CESM(1) Large Ensemble (Kay et al., 2015), hereafter referred to as "CESM1-LE", which consists of 40 members initialised from with random round-off error from a single ensemble member in 1920 and freely evolving thereafter. Each ensemble member is performed with identical external forcing, following the CMIP5 protocol for the 1920-2005 historical period and the representative concentration pathway 8.5 (RCP 8.5) over the period 2006-2100. The second large ensemble dataset is the MPI Grand Ensemble (Maher et al., 2019), hereafter referred to as "MPI-GE", which is similar to CESM1-LE but uses the MPI Earth System Model and consists of 100 members starting in 1850, each initialised from a different initial conditions taken from a long pre-industiral control simulation. MPI-GE is integrated through to 2099 using various CMIP5 forcing scenarios but here we use the RCP 8.5

data to compare with CESM1-LE. We only use 99 members of the MPI-GE that had all of the variables used here available at the time of carrying out the analysis. For both datasets we use data over the period 1920-2060, which is covered by both large ensemble datasets. The near-term ($\approx$ 1-40 years) period is the primary period of interest of the EUCP project.

Observational data for surface air temperature and precipitation is taken from the CRU-TS v4.01 gridded surface dataset (Harris et al., 2014). The observational sea-level pressure (SLP) data is taken from the HadSLP2 dataset (Allan and Ansell, 2006) for the results presented below, however, we tested the sensitivity to the choice of observational SLP dataset by using the 20th Century Reanalysis v3 (20CR; Compo et al., 2011). The results were generally very similar regardless of the observational datasets, however, some of the differences between the observational dataset are highlighted in section 3.5.

The data used for out-of-sample verification was taken from the CMIP5 archive (Taylor et al., 2012). We take the first ensemble member for the 39 models that cover the 1920-2060 period for the historical (up to 2005) and RCP 8.5 (from 2006) scenarios. The CESM1-LE has a $1°x1°$ degree horizontal resolution in the atmosphere (with 30 vertical levels), which is generally comparable or higher resolution than the models in the CMIP5 ensemble. The MPI-GE has a comparatively low T63 spectral resolution (equivalent ot around a $2°$ degree horizontal resolution), with 40 levels in the vertical. Data from the CMIP5 models, MPI-GE ensemble and the observational datasets were regridded to the same grid as the CESM1-LE dataset prior to the analysis. Tests on a small subset of the results showed that the results were not sensitive to the regridding procedure.

We analyse the evolution and projections of surface-air temperature (referred to as temperature hereafter) and precipitation over the three European SREX regions (Field et al., 2012). These are the Northern Europe, Central Europe and Mediterranean regions, which are shown in Figure 1 and will be hereafter referred to as NEUR, CEUR and MED, respectively. Our analysis focuses on projections of seasonal mean climate for European summer (defined as the June-July-August average) and winter (defined as the December-January-February average).

## 2.2 Verification metrics

The impact of the calibration is assessed through a series of verification metrics. The root-mean square error (RMS error) is a simple measure of the accuracy of the ensemble mean prediction. In addition, the spread of the ensemble is also calculated, which is defined as the square root of the mean ensemble variance over the verification period (e.g. Fortin et al., 2014). By calculating the RMS error and spread we are able to estimate the reliability of the ensemble by calculating the spread/error ratio, which for a perfectly reliable ensemble will be equal to one (e.g. Jolliffe and Stephenson, 2012). A spread/error ratio greater than one indicates an underconfident ensemble, whereas a spread/error ratio less than one indicates an overconfident ensemble. The final metric that we will consider is the continuous rank probability score (CRPS), which is a probabilistic measure of forecast accuracy that is based on the cumulative probability distribution (e.g. Hersbach, 2000; Wilks, 2011; Bröcker, 2012). The CRPS measures where the verification data point lies with respect to the underlying ensemble and is higher when the verification data are further from the centre of the ensemble. As such, a lower CRPS value represents a more skillful probabilistic forecast.

## 2.3 Ensemble calibration methods

We will assess the effectiveness of calibrating ensemble climate projections using a series of different calibration techniques, which are outlined in this section. The calibrations are performed separately for each region and season, on annually-resolved indices.

### 2.3.1 Uncalibrated ensemble

The benchmark for the calibration methods is the uncalibrated ensemble. Here we use the term uncalibrated ensemble to refer to an ensemble that has been bias corrected by removing the mean value over a particular reference period. Of course, this is not strictly an uncalibrated ensemble but this is the most common way that climate projections are presented in the literature (e.g. Hawkins and Sutton, 2016). In the analysis that follows the reference period is always the same as for the corresponding calibration methods, which is generally the observational period 1920-2016 in the following analysis.

### 2.3.2 Variance inflation (VINF)

One calibration method that we will test is "Variance inflation", hereafter referred to as VINF, following Doblas-Reyes et al. (2005). For each uncalibrated ensemble member, $X_{\text{uncalib}}$, VINF adjusts the ensemble mean signal, $X_{\text{m}}$, and anomaly with respect to the ensemble mean, $X_{\text{ens-anom}}$, from the uncalibrated ensemble. The uncalibrated ensemble can be expressed in these terms as

$$X_{\text{uncalib}}(t, e) = X_{\text{m}}(t) + X_{\text{ens-anom}}(t, e). \tag{1}$$

Here $t$ and $e$ indicate dependence on time and ensemble member, respectively. The VINF method produces a calibrated ensemble, $X_{\text{calib}}$, through the following scaling

$$X_{\text{calib}}(t, e) = \alpha X_{\text{m}}(t) + \beta X_{\text{ens-anom}}(t, e). \tag{2}$$

The scaling variables $\alpha$ and $\beta$ are calculated as

$$\alpha = \rho \frac{s_{\text{r}}}{s_{\text{m}}}; \tag{3}$$

$$\beta = \sqrt{1 - \rho^2} \frac{s_{\text{r}}}{s_{\text{uncalib}}}; \tag{4}$$

where $s_{\text{r}}$ is the standard deviation of the reference (or observational) data that is being calibrated towards, $s_{\text{m}}$ is the standard deviation of the ensemble mean, $s_{\text{uncalib}}$ is the square-root of the mean variance of the uncalibrated ensemble members, and $\rho$ is the correlation between the ensemble mean signal and the reference dataset over the calibration period. Where $\rho$ is less than zero and there is no skillful correlation between the ensemble and the reference dataset, we set $\rho$ to be zero in the calibration. VINF scales the signal and ensemble spread but maintains the underlying correlation and ensemble distribution, rather than fitting a parametric distribution as in the following methods.

### 2.3.3 Ensemble Model Output Statistics (EMOS)

The next calibration method is the "Ensemble Model Output Statistics" approach, hereafter referred to as EMOS (Gneiting et al., 2005). The EMOS method has widely been applied to the output of ensemble prediction systems for medium-range and seasonal forecasts. EMOS involves fitting a parametric distribution to the underlying data, such that the uncalibrated is expressed as

$$X_{\text{uncalib}}(t) = X_{\text{m}}(t) + \epsilon_{\text{uncalib}}(t), \qquad \epsilon_{\text{uncalib}}(t) = N[0, s^2(t)]; \tag{5}$$

and the calibrated ensemble is expressed as

$$X_{\text{calib}}(t) = bX_{\text{m}}(t) + \epsilon_{\text{calib}}(t), \qquad \epsilon_{\text{calib}}(t) = N[0, c + ds^2(t)]; \tag{6}$$

where $N[\mu, \sigma^2]$ is a Gaussian distribution with mean $\mu$ and variance $\sigma^2$ and $s^2$ is the time-dependent variance across the ensemble. The coefficients $b, c$, and $d$ are found using numerical methods to minimise the CRPS over the calibration period Gneiting et al. (2007). The coefficients $b, c$, and $d$ are constrained to be non-negative values. The EMOS technique is arguably the most general method we will test because it allows for meaningful differences in spread across the ensemble at different times (i.e. the coefficient $d$), so is sometimes referred to as "Nonhomogenous Gaussian Regression" (e.g. Wilks, 2006; Tippett and Barnston, 2008). EMOS represents a simplification over the VINF method because the the ensemble distribution is parameterised as Gaussian. In this study, the EMOS technique is used to produce 1000 sampled ensemble members in the ensemble projection. To avoid overfitting to the observations when producing the ensembles and to include some measure of sampling uncertainty in the parameter fitting process, the EMOS method is applied to randomly resampled years (with replacement) from the calibration period, to produce 1000 valid combinations of the coefficients $b, c$, and $d$. These combinations are used to produce the 1000 sampled ensemble members used to produce the calibrated projection.

### 2.3.4 Homogenous Gaussian Regression (HGR)

The third calibration method that we test is "Homogenous Gaussian Regression", hereafter referred to as HGR. The HGR method is a simplified version of EMOS, in which the calibrated variance is constant in time and is expressed as

$$X_{\text{calib}}(t) = bX_{\text{m}}(t) + \epsilon_{\text{calib}}(t), \qquad \epsilon_{\text{calib}}(t) = N[0, c]. \tag{7}$$

Effectively, this method assumes that there is no information in the time variation of the ensemble spread. The coefficients $b$ and $c$ are found as in EMOS and are constrained to be greater than or equal to zero.

### 2.4 Dynamical decomposition of climate anomalies

In this study we will test calibrating the full variables as well as calibrating dynamically decomposed variables. The dynamical decomposition aims to express variables - surface air temperature and precipitation in this case - as a dynamical and residual component. The rationale for testing this on the calibration methods is that they may be fitting a thermodynamic signal in the

ensemble to something that is dynamically driven in the reference (or observational) data and therefore conflating different mechanisms. Dynamical decomposition has previously been used to understand observed large-scale climate variability on decadal timescales where there is a contribution from the thermodynamic climate change signal and large-scale circulation anomalies (e.g. Cattiaux et al., 2010; Wallace et al., 2012; Deser et al., 2016; Guo et al., 2019). The dynamical decomposition

splits the variables at each grid-point over Europe into *FULL = DYNAMICAL + RESIDUAL*.

The dynamical component was calculated for all model ensemble members, CMIP5 models and observations following the analog method of Deser et al. (2016). The method here is exactly the same as that used in O'Reilly et al. (2017), which provides full details. In this method, sea-level pressure (SLP) anomaly fields for each month are fitted using other SLP anomaly fields from the corrensponding month from other years over the reference/observational period (1920-2016). This regression

fit yields weights which are then used to compute the associated dynamical surface temperature or precipitation anomaly. Each field can then be separated into a dynamical and residual component. An example of the dynamical decomposition of the CESM1-LE projection into dynamical and residual components is shown in Figure S1 (and also in the example calibration schematic in Figure 3). The regional dynamical and residual timeseries were calibrated using the above techniques towards the corresponding dynamical and residual timeseries from the target dataset (i.e. CMIP5 or observations over the period 1920-

2016). The calibrated dynamical and residual timeseries are then combined to give a full calibrated ensemble projection, further detail is provided in the following section. Results from the calibrated dynamical decomposition are shown later in the paper for the HGR method and referred to as HGR-decomp.

## 3   Results

### 3.1   An example ensemble calibration

Before we begin our analysis, it is useful to motivate our approach by briefly describing an example calibration. A synthetic, randomly-generated 100 member ensemble is shown in Figure 2, alongside a synthetic observational index. There is a large spread across the ensemble, with the reference frequently lying close to the ensemble mean. The lower panel of Figure 2 shows the ensemble calibrated towards the reference data using the VINF method. The VINF method scales the ensemble mean and spread to make the ensemble reliable in a probabilistic sense. The improvement of the calibrated ensemble is clear from the

reduction in error and CRPS, which is also shown in Figure 2. Also, it is important to note that calibrated ensemble is perfectly reliable over the reference period, as indicated by the spread/error being equal to 1 after calibration. The EMOS and HGR methods would have yielded almost identical results for this synthetic ensemble.

It is clear from the example shown in Figure 2 that it is trivial to calibrate an ensemble to known data such that it is perfectly reliable. Of more interest here is whether calibrating to observed data can improve the accuracy and reliability of a prediction

*outside* of the reference period used for the calibration.

### 3.2 Comparing calibration techniques using an "imperfect model" test

Our aim in this study is to test how calibrating large ensemble projections using observations will influence the accuracy and reliability of the projections. The common period of the large ensembles and observations used in this study is 1920-2016, so we can in principle calibrate the ensembles using this period. However, we cannot test how effective this calibration is in the future, out-of-sample period. To examine the performance of the calibration we employ an "imperfect model" test, using 39 CMIP5 models. In this test, the large ensemble dataset is calibrated to each of these 39 models over the reference period, 1920-2016. The future period from the CMIP5 realisation is then used to analyse the impact of the calibration by verifying against calibrated large-ensemble in the out-of-sample period 2017-2060. We refer to this as an imperfect model test because, in this approach, the large ensemble calibration is tested mostly on simulations from different climate models. This is a strength of the imperfect model test, as the observations can, in some sense, be considered an out of sample test. In addition to the 1920-2016 calibration period, we also tested the calibration over shorter periods (some examples are shown in Figure S8 of the Supplementary Material). Overall, the calibration was found to perform better over the longer periods, so in this study we focus on the results of the calibration on the longest available common period (i.e. 1920-2016).

An example of calibrating a large ensemble projection to a CMIP5 model index is shown in the left hand column in Figure 3. In this example, the uncalibrated CESM1-LE data for CEUR summer temperature is shown in red, along with the same index from one of the CMIP5 models over the reference period (1920-2016). The model is calibrated towards data from the CMIP5 model realisation over the reference period. Following the calibration step, the CMIP5 data from the future period (2017-2060), which was withheld prior to the calibration, is used to verify the uncalibrated and calibrated large-ensemble projections using each individual year in the verification period. The verification is performed on 44 pairs of probabilistic predictions and validation data points from this future period (2017-2060). From each of the CMIP5 models, we can therefore calculate verification statistics (i.e. RMSE, Spread/Error, CRPS). The process is then repeated for each of the 39 CMIP5 models and the distribution of these verification statistics is presented in the results that follow. We performed this analysis for each of the calibration methods using both the CESM1-LE and MPI-GE datasets. The analysis for both temperature and precipitation, for summer and winter, and over all three European regions is presented and discussed below.

The verification statistics for the CESM1-LE summer temperature for the uncalibrated ensemble and the three calibration methods are shown in Figure 4. The distribution of the verification statistics over the 39 models is shown, with the horizontal lines indicating the median of the distribution. The black crosses indicate where the verification of the calibrated ensemble is significantly better than the verification of the uncalibrated ensemble at the 90% confidence level, calculated using the non-parametric Mann-Whitney U-test (e.g. Wilks, 2011). For the summer temperature over all three regions, all of the calibration methods significantly lower the RMS error of the ensemble projection compared with the uncalibrated ensemble. The calibration methods generally perform similarly, acting to typically reduce the spread of the uncalibrated ensemble and narrowing the range of the spread/error ratios in the verification compared to the uncalibrated ensemble. There is significant improvement in reliability, indicated by the spread/error relationship, for the CEUR region with all three calibration methods. The CRPS is significantly lower for all of the calibration methods in all regions, demonstrating that the calibrations are improving the

probabilistic predictions of summer temperature by the CESM1-LE ensemble in the out-of-sample future period. An important point to note is that, despite not being a significant improvement for all the verification metrics shown in Figure 4, none of the calibration methods ever has a *significantly* negative impact on the projections.

We also performed the same testing described above for the CESM1-LE on the on the MPI-GE. The verification measures for the MPI-GE summer temperature are shown in Figure S2. The performance of the calibration methods on the MPI-GE summer temperature is qualitatively similar to that for the CESM1-LE (shown in Figure 4). The calibration methods in general improve the out-of-sample verification statistics, resulting in a more accurate and reliable projection over the three European regions compared to the uncalibrated ensemble. Again, there is a particularly notable improvement for the CEUR region, as with the CESM1-LE data (i.e. Figure 4). For the other regions there is an improvement over the uncalibrated ensemble but this is not significant for any of the calibration methods, or for any of the verification measures. Nonetheless, as in the CESM1-LE data, none of the calibration methods displays a significantly negative impact on the projections.

The comparison of the ensemble calibration methods for the summer temperature suggests that there is no significant difference between the performance of the different methods, for both of the large ensembles (i.e. Figure 4 and S2). Analysis of the equivalent figures for precipitation (Figures S3 S4), as well as for the the winter season (not shown), also demonstrate a reasonably consistent performance between the calibration methods. The similarity of the performance of the calibration methods indicate that the extra information included in the VINF and EMOS calibrations, compared with the HGR calibration, is not important to the performance. Therefore, we will focus on the simplest method of the three, HGR, for the analysis that follows.

The out-of-sample verification results for the HGR method for temperature and precipitation for both summer and winter seasons for the calibrated CESM1-LE projections are shown in Figure 5 (note that the red and orange data in the first column are the same as those shown in Figure 4). The equivalent verification plot for the MPI-GE dataset is shown in Figure S2, and the results are, generally, qualitatively similar to those shown for the CESM1-LE dataset in Figure 5. The improvement for the winter temperature in terms of RMS error is not as clear as in the summer season but the reliability of the ensemble projections are improved significantly over the NEUR region. There is also some improvement for the precipitation projections in some regions, particularly in terms of the spread/error ratios of the regional precipitation projections. The spread of the uncalibrated CESM1-LE data seems to be larger than is appropriate for the targeted indices, particularly for precipitation, which is evident in the general reduction in spread in the calibrated ensemble. The spread/error ratios of the calibrated ensembles are consistently close to one, this is a particularly notable improvement for the uncalibrated ensembles over the NEUR region, which are generally underconfident prior to calibration. For some other regions, there is a smaller improvement or no noticeable difference. Crucially, the influence of the calibration on the spread/error is not significantly negative for any of the the variables regions or seasons, indicating that the calibration generally improves the reliability of the projections. The only verification statistic where the calibrated ensemble performs significantly worse than the uncalibrated ensemble is the RMS error for the NEUR winter precipitation in the CESM1-LE dataset (Figure 5). Whilst it is only one of the verification measures performed across both the CESM1-LE and MPI-GE datasets, it is a concern because it reduces how much confidence we can have in applying the calibration using observations.

### 3.3 Examining calibration using dynamically decomposed variables

One potential problem with the calibration methods examined in the previous section is that they are calibrated towards a single (observational) index. The implicit assumption with this calibration approach is that the forced signal in the large ensembles is scaled based on the co-varying signal in the reference/observational index. However, we might expect the forced climate change signal to be largely thermodynamic in nature rather than being driven by changes in large-scale circulation, particularly for temperature. It is possible therefore that when fitting the calibration of the ensemble to the reference, there is an incorrect conflation of, for example, the forced thermodynamic response with an circulation driven signal associated with internal variability in the reference index. To account for this potential shortcoming in the calibration method we used a dynamical decomposition method (as outlined above in section 2.4) to split the model and observations datasets into a *DYNAMICAL* component, associated with large-scale circulation anomalies, and a *RESIDUAL* component, which can often be interpreted as a thermodynamic component.

An example of the dynamical decomposition, applied to summertime projections for the CEUR region in the CESM1-LE dataset, is shown in Figure 3 (and also Figure S1). In this example, the future temperature response is largely associated with the residual, representing the local thermodynamic response to increase greenhouse gas concentrations. There is also some dynamical contribution to the signal but this also contributes to the uncertainty in the overall ensemble projection. In contrast, there is a much weaker signal in future precipitation changes, and the modest drying signal that is projected seems to be mostly due to dynamical changes.

We will now examine how calibrating the dynamically decomposed parts of the ensemble projection (e.g. the *DYNAMICAL* and *RESIDUAL* components) separately, against the respective decomposed parts of the reference indices, before recombining affects the ensemble calibration performance. A demonstration of this process applied to one of the CESM1-LE projection is shown in Figure 3. The large ensemble projection and reference dataset are both separated into dynamical and residual components. These are then calibrated separately, which in this particular example reduces the dynamical signal substantially. Next, the calibrated decomposed projections are recombined to produce the total calibrated projection. This total calibrated projection is then used to calculate verification statistics, in the same way as for the full calibration techniques examined previously. We use the HGR method to perform the calibration on the dynamically decomposed data, and refer to this method as "HGR-decomp" hereafter.

Verification results for the HGR-decomp methods are shown for both temperature and precipitation and for all regions in Figure 5, alongside the HGR verification results (with the equivalent verification for the MPI-GE shown in Figure S5) As with the HGR verification, the crosses/circles indicate where the verification statistics of the HGR-decomp calibrated ensemble are significantly better/worse than the uncalibrated ensemble. The HGR-decomp calibration generally performs better than the uncalibrated ensemble, and for none of the verification measures does the HGR-decomp calibration perform significantly worse than the uncalibrated ensemble. This is in contrast with the HGR calibration method, for which there is a significant increase in the RMS error for the wintertime precipitation in the NEUR region.

To formally compare the HGR-decomp and HGR, we assessed the significance of the difference in the verification measures of the two methods. In Figure 5, the black boxes indicate where either of the calibration methods is found to be significantly better than the other, at the 90% level (based on a Mann-Whitney U-test). The only statistically significant differences are seen for the spread/error verification, where four of the regions/variables are significantly better for the HGR-decomp method applied to the CESM1-LE dataset. In contrast, none of the verification measures for any of the regions/variables are significantly worse for the HGR-decomp method. The HGR-decomp method also performs better for the calibrated MPI-GE indices (Figure S5), albeit with a lower level of significance. Specifically, in ten of the twelve total regions/variables verified for the MPI-GE dataset, HGR-decomp calibraton is found to be more reliable in terms of spread/error than the HGR calibration.

Overall, the HGR-decomp method is found to be an improvement over the HGR method, and very clearly outperforms the uncalibrated ensembles. The improvement of the HGR-decomp method over the HGR method is clearest in the reliability of the projection, as measured in terms of spread/error. The spread/error is consistently higher in the HGR-decomp calibration, primarily due to the spread, which is consistently larger in the HGR-decomp calibrated ensemble. Calibrating on the dynamical and residual components separately has the effect of increasing the overall spread, likely because the method avoids fitting a forced thermodynamical or dynamical signal in the ensemble towards a forced or internal variability of a different origin in the reference index. Examining the verification of the HGR calibrated *DYNAMICAL* and *RESIDUAL* components separately reveals that the spread/error of the *DYNAMICAL* components of the ensemble are particularly well calibrated (not shown). In comparison with the HGR-decomp method, the HGR method generally has a lower spread, which in many cases results in projections that have a spread/error ratio lower than one and are less reliable than for the HGR-decomp method. In this sense, the HGR method appears to be slightly "over-fitting" the ensemble to the reference period, resulting in a consistently over-confident ensemble projection.

### 3.4 Examining the impact of calibration on projections of future climatologies

To assess how the calibration influences the projections of average European climate during the mid-21st century period, we will examine projections of the mean 2041-2060 climate. Until this point we have focused on verifying the yearly projections of each season over the out-of-sample period 2017-2060, which gives a verification measure for each CMIP5 model (e.g. as shown in Figures 4 & 5). We also need to verify the out-of-sample projections for the 2041-2060 means. However, since there is only a single verification point for the climatology in each of the CMIP5 models, we instead need to to combine the single measurements to produce one verification score across all the models. To estimate the uncertainty of these verification measures, we perform a bootstrap resampling over the 39 CMIP5 model projection/verification pairs. Verification results for the 2041-2060 climatologies for both the CESM1-LE and MPI-GE are shown in Figure 6.

The HGR-decomp calibration tends to improve the projected 2041-2060 climatology of temperature in both seasons and ensembles, but especially summer. This is a particular improvement during the summer, in both the accuracy (i.e. RMS error) and reliability (i.e. spread/error) of the out-of-sample verification. The calibrated summer temperature projections are more reliable in all three European regions in the CESM1-LE and MPI-GE ensembles but all tend to somewhat overconfident. The winter temperature shows less obvious improvement in terms of RMS error of the calibrated projections, but the reliability is

significantly improved for all the regions in both ensembles, but again, the calibrated CESM1-LE data is slightly more reliable. There is less improvement for precipitation projections than seen for the temperature projections. For the summer precipitation, there are modest but significant improvements in some regions in terms of the RMS error but the reliability is more mixed, with the calibration actually worsening the reliability in the MED region for the CESM1-LE dataset. The calibration has the least influence on the 2041-2060 climatology of precipitation, acting to worsen the RMS error in some instances but also to modestly improve the reliability.

Overall, the verification of the projected 2041-2060 climatologies in the imperfect model tests indicate that the HGR-decomp calibration acts to generally improve the accuracy and reliability of the projections. The calibrated temperature projections perform better than the calibrated precipitation projections. It is notable however, that the out-of-sample verification for the 2041-2060 climatologies do not generally seem to perform as well as the calibration for the yearly projections examined in the previous sections. There are several possible factors contributing to this. The first is that when we examine the performance of the calibration on the yearly projections, the beginning of the 2017-2060 verification is found to be more accurate and reliable than the latter period, as the forced signal in the ensemble diverges from the observations to which it is calibrated. This is demonstrated clearly when the verification is applied to different future periods (specifically 2021-2040, 2041-2060 and 2061-2080; see Figure S6). We find that the accuracy and reliability clearly deteriorate as the target period moves further into the future, indicating that the HGR-decomp calibration method is less appropriate for periods further into the future. Another reason is that much of the increased reliability in the yearly projections stems from calibrating the (unpredictable) internal variability in the ensemble to the target index, but in the 20-year climatology there is a much smaller contribution of this internal variability.

## 3.5 Calibrating large ensembles to observations and assessing the impact on future climate projections

The imperfect model tests in the previous sections demonstrate that the calibration methods generally act to improve future projections in a out-of-sample verification. In particular, the HGR-decomp method is a categorical improvement over the un-calibrated ensembles in the imperfect model analysis using the CMIP5 ensemble, as described in the previous sections. On the basis of this analysis, we will now apply the HGR-decomp calibration method to the large ensembles, using the observational indices of temperature and precipitation to calibrate against.

The calibrated CESM1-LE projections for the summer temperature and precipitation are shown in Figure 7. Based on the imperfect model tests we expect the calibrated summer CESM1-LE to represent the most accurate and reliable projection out of all of the ensemble/variable/season combinations tested. For the summer temperature projections, calibrating the CESM1-LE against the observations over the reference period, the rate of warming until 2060 is reduced by varying amounts. There are also small changes in spread, perhaps most notable for the NEUR region, with the calibration method acting to increase the uncertainty in the future projections. For the precipitation, the signal in CESM1-LE projection is much weaker with respect to the inter-annual variability. In the projections shown here the calibration has a notable impact on the future projections, acting to weaken the drying projected in the CEUR region and adjusting the ensemble uncertainty in all the regions.

The calibrated summer projections from the MPI-GE are fairly similar to the CESM1-LE, with the ensemble medians of the MPI-GE also plotted in Figure 7 for comparison (the full ensemble projection plots are shown in Figure S7). The warming in the NEUR region is reduced over the 2017-2060 period and the uncertainty is increased markedly. The calibration technique makes smaller adjustments to the summer temperature projections in the CEUR and MED regions, which may be because the uncalibrated ensemble already does a reasonable job of capturing the warming variability seen during the observational period. In the projections of summer precipitation in the MPI-GE dataset, there is a fairly strong future drying signal in both the CEUR and MED regions that is greatly reduced by the calibration. Interestingly a similar result is seen in the time-slice experiments of Matsueda et al. (2016) when calibrated using the results of seasonal hindcast experiments, which tends to reduce the drying in the MED region. In the calibration shown here, this seems to be because the MPI-GE has a drying trend over the whole observational period in these regions, which is dynamical in origin and is not seen clearly in the observations. Based on the imperfect model tests, however, we have less confidence in the performance of the calibrated ensembles for precipitation.

We also tested the observational calibration using data decomposed using the 20CR SLP, rather than the HadSLP2 SLP data (Figure S9). The results are generally insensitive to the coice of SLP dataset. One exception is the MED summer temperatures, for which the calibration amplifies future warming. In this instance the DYNAMICAL component of the decomposition when using 20CR SLP accounts for substantially less of the observed variance than in HadSLP2 (Table S1) and the decompositions are also substantially different (Table S2). This indicates that for this season and region the 20CR data is not capturing what seems to be a clearer dynamical signal in the HadSLP2 dataset and, as a result, is perhaps less dependable. On the whole though, the results are largely insensitive to the choice of SLP dataset.

To consider whether the imperfect model testing is really a useful indication of the performance of the observational calibration, it is of interest to compare the fit parameters of the HGR-decomp calibration. The parameters $b$ and $c$ from equation 7 are plotted in Figure S10. The observed scaling parameters, $b$ and $c$, generally lie within the range of values used to calibrate CMIP5 models in the previous sections. In some cases the parameters lie outside the CMIP5 model ensemble but this is not systematic, so there is no clear reason to expect the efficacy of the calibration to be very different when applied to the observational data.

The projected changes in the 2041-2060 climatologies, compared with the present day 1995-2014 reference period, are shown in Figure 8. Here, we have plotted both the uncalibrated and (HGR-decomp) calibrated climatological changes for both the CESM1-LE and MPI-GE datasets. An interesting feature of these projected changes is that for many of them, the calibrated ensembles are more consistent with one another than their uncalibrated counterparts. This is perhaps most clear for the summer temperature changes in all the European regions, particularly NEUR and CEUR, in which there is a difference of over $1°K$ in the mean changes of the uncalibrated projections and with no overlap in the probability distributions. After the calibration is applied, the projected mean changes are closer to one another, with considerable overlap in their probability distributions. The calibration acts to make the projections more consistent for most of the variables and regions, which is reassuring as this implies that the observations are having a strong impact on the initial uncalibrated ensembles that are themselves often very different.

Another feature of the calibrations influence on the future climatologies is that it fairly consistently acts to increase the uncertainty of the projections, with respect to the uncalibrated ensembles. This is most clear for the projections of future temperature over Europe, where the imperfect model tests indicate that the calibration has a large impact on the reliability of the projections (e.g. Figure 6), suggesting that the broader calibrated distribution is reasonable and is likely to be a better future projection. It is interesting to note that the calibrated CESM1-LE projection has a wider spread than in the calibrated MPI-GE projection for many of the projected temperature indices, which may be related to particular trend biases in the CESM1-LE (e.g. McKinnon and Deser, 2018). In the imperfect model tests, shown in Figure 6, the calibrated temperature projections for the CESM1-LE dataset are consistently more reliable (in terms of spread/error) than for the MPI-GE dataset. The calibrated MPI-GE projections were more underconfident in the out-of-sample verification, indicating that we should have more confidence in the broader calibrated CESM1-LE projections for future temperature changes.

In a recent paper, Brunner et al. (2020) compared several different methods of model weighting and constraining climate projections for the European summer season using multi-model ensembles over the same 2041-2060 period under the RCP 8.5 scenario. The HGR-decomp method generally predicts lower levels of warming for European summer than the CMIP5-based model weighting/constraining methods but much of the distributions of the projected changes overlap. It is notable that the HGR-decomp method can project changes that are outside of the uncalibrated distribution, which is clearly not the case for the model weighting/constraining methods (see, for example, Figure 2 of (Brunner et al., 2020). This feature in particular sets the HGR-decomp method apart from these other techniques, whether this is for better or worse though, is not clear.

## 4    Conclusions

In this study we have examined methods of calibrating regional climate projections from large single model ensembles. The three calibration methods tested here - VINF, EMOS and HGR - are more commonly used for initialised forecasts from weeks up to seasonal timescales. Here we applied these calibration techniques to ensemble climate projections, fitting seasonal ensemble data to observations over a reference period (1920-2016). The calibration methods act to scale the ensemble signal and spread so as to optimize the fit over the reference period. The three calibration methods display similar performance, all generally improving the out-of-sample projections in comparison to the uncalibrated ensemble. The simplest of the calibration methods, HGR, includes no variability of the ensemble spread and effectively discards any information that may be contained in the year-to-year variability of the spread in the raw ensemble. Based on the performance of the HGR method, we can conclude that the information in the year-to-year changes in the ensemble spread is not important enough, at least in the large ensembles examined in this study, to have a meaningful influence on the ensemble calibration.

We also tested calibrating the variables after they had been subjected to a dynamical decomposition. In this method, all variables were separated into *DYNAMICAL* and *RESIDUAL* components using information of the large-scale circulation, calibrated separately using the corresponding reference indices and then recombined to produce the final calibrated ensemble. The results from the out-of-sample verification of the HGR-decomp calibrations demonstrate a small but noticeable improvement over the HGR method, particularly in terms of the reliability. The HGR calibrated ensembles have a tendency to be overconfident

for their future projections and this seems to be due to an apparent over-fitting to variability in the reference period, which is found to be alleviated to some extent by calibrating the *DYNAMICAL* and *RESIDUAL* components of the ensemble separately. Therefore, the HGR-decomp calibration was chosen as the best method to apply to the observational reference data.

The HGR-decomp calibration method was also found to improve the projections of 20-year climatologies during the mid-21st century (i.e. 2041-2060). The accuracy and reliability of the projections improve in the calibrated ensemble, when subject to the imperfect model tests. The performance of the calibration was substantial for the temperature projections but for precipitation the improvement is much more modest, or even absent in some instances. Whilst both datasets demonstrate an improvement due to calibrations, it is interesting that the CESM1-LE dataset seems to perform better than the MPI-GE, particularly in terms of the reliability of the future projections. Perhaps it is not too surprising that one ensemble would be found to be better than another when subjected to this type of calibration. For example, if we had a third ensemble that we knew was a much worse representation of the climate system, we might expect the calibration to improve the projection in this ensemble but we would not expect this calibrated ensemble to outperform the other large ensembles. In this sense, the calibration approach taken here is clearly not a panacea for all ensemble projections and ultimately, the accuracy and reliability of the calibrated ensemble projection would expected to depend on the raw ensemble projection.

We then proceeded to apply HGR-decomp method to each of the large ensembles using the observations as a reference, over the period 1920-2016. Based on the imperfect model testing we expect that the calibrated ensemble projection provides a more accurate and certainly a more reliable probabilistic projection of European climate over the next 40-50 years. In both the CESM1-LE and MPI-GE datasets the projected increase in European temperatures is generally smaller in the calibrated ensembles compared to the uncalibrated ensembles. The calibrated projections are notably more consistent with one another than the respective calibrated projections, indicating that the calibration with observations is having a consistent and substantial influence on the future projections of European climate. For the example of European temperatures, the best estimates for the summer temperature change for the period 2041-2060 (from 1995-2014) is projected to be about $2°C$ for CEUR and MED regions and $1.3°C$ for the NEUR region. Each of these is associated with a substantial ensemble spread, however, reflecting the increased uncertainty (or larger ensemble spread) added by the calibration to provide a more reliable projection.

The overall effectiveness of the calibration seems to stem from some key characteristics of the ensemble and reference datasets. The calibration performs well where there is a reasonably strong signal in the ensemble that is also present to some extent in the reference data, as is the case for the temperature indices. In these instances, the signal is scaled and an ensemble spread is added to represent the appropriate estimate of internal variability, much of which is associated with large-scale circulation variability. For precipitation, where there is no clear signal over the reference period in the observations for the specific regions and seasons analysed here (and in many of the CMIP5 models), any future changes projected are difficult to scale over this reference period. In effect, the calibration then adds value by correcting (mostly by inflating) the ensemble spread. This calibration method could therefore reasonably be applied to many other regions and variables where there is an emerging forced signal in response to external forcing. The calibration can also be applied to smaller spatial scales but as the scales become smaller, the forced signal generally becomes weaker relative to the internal variability, so the calibration

will tend to become somewhat less effective. Nonetheless, the calibration has also demonstrated some utility for temperature projections on 2.5° grid-boxes (as included in Brunner et al. (2020)).

One novel aspect of this study that is particularly worth emphasising is the imperfect model testing approach. Previous studies have typically used multi-model ensembles to constrain future projections and some in particular have used a "leave-one-out" perfect model approach to examine the effectiveness of these methods (e.g. Knutti et al., 2017; Brunner et al., 2019). However, this leave-one-out approach is often used to tune particular parameters in the methodology, such as the performance weighting parameter in Brunner et al. (2019). The use of the leave-one-out approach to tune the method is certainly well justified. However, this does reduce the power of subsequently re-using this approach to verify the accuracy of the constraining method, which may result in over-fitting or an over-estimation of the added-value of the constraint. The imperfect model approach we have used in this study is less susceptible to this type of over-fitting as the data used for the verification are kept separate from the underlying large ensembles throughout, only being used to compare the efficacy of the different methods. The imperfect model approach to testing is therefore an advantageous approach, regardless the particular calibration or model weighting that is being subjected to the testing.

As well as being applied to other datasets and regions, this calibration method can also be applied to initialised decadal forecasts. Decadal predictions exhibit skill in some aspects out to 10-years (e.g. Doblas-Reyes et al., 2013; Smith et al., 2019) and a recent study has demonstrated that constraining climate projections using initialised decadal predictions can improve the accuracy of projections in some cases (Befort et al., 2020), which is an exciting proposition for improving climate prediction. Given that these calibration methods have been shown to be effective when applied to initialised decadal forecasts, if calibration also proves effective for projections beyond 10 years this would present an opportunity to merge the calibrated decadal predictions with calibrated large ensemble climate projections.

Previous studies have examined how similar calibration methods to those examined in this paper can improve multi-year forecasts (e.g. Sansom et al., 2016; Pasternack et al., 2018). It would be of particular interest to examine how calibrated decadal predictions could be combined or merged with these calibrated projections. The CESM1-LE dataset analysed in this study has an initialised counterpart, namely the Decadal Prediction Large Ensemble (Yeager et al., 2018), and testing how to combine data from these different ensembles to produce a merged calibrated set of climate predictions would potentially be an exciting extension to the present study.

*Acknowledgements.* This study is part of the European Climate Prediction system project (EUCP). The EUCP project is funded by the European Commission through the Horizon 2020 Programme for Research and Innovation: Grant Agreement 776613. We are thankful for the comments and suggestions of Francisco Doblas Reyes and three anonymous reviewers that helped us to improve our study.

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

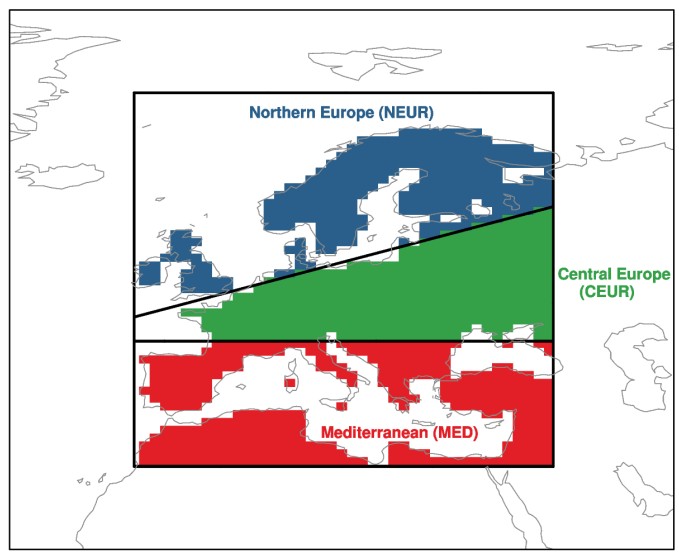

**Figure 1.** The SREX regions over which area-averaged projections and observations are analysed in this study, following Field et al. (2012).

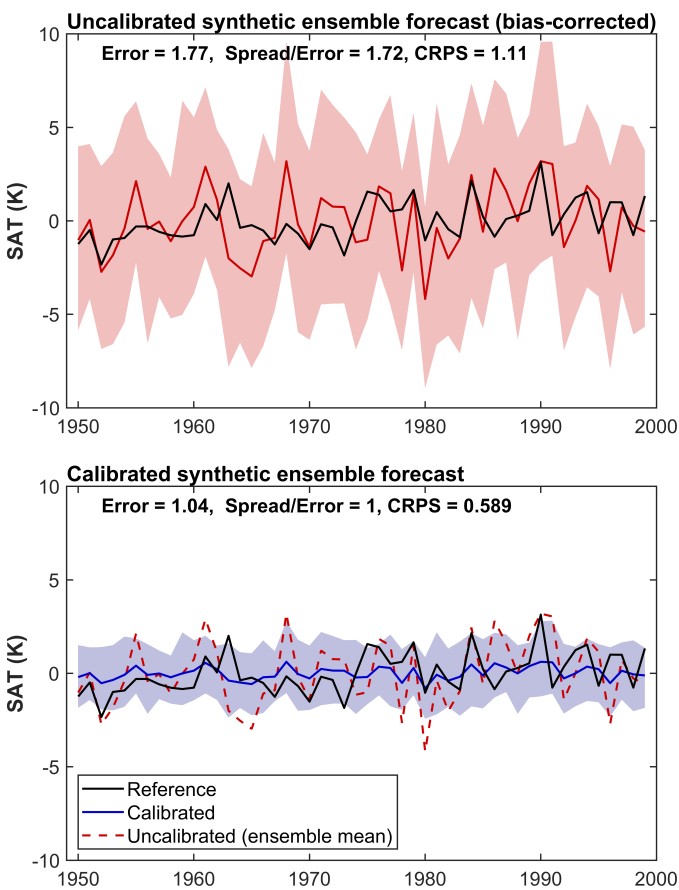

**Figure 2.** (Top) Synthetic data for an example (bias-corrected) ensemble temperature evolution is shown for the ensemble mean in red and 90% ensemble range (shaded), along with the synthetic (observational) reference index in black. (Bottom) The synthetic ensemble calibrated using the variance inflation method to match the reference dataset shown in blue with the raw ensemble mean shown in dashed red. The RMS Error, spread/error and CRPS calculated for the raw ensemble and the calibrated ensemble are all shown.

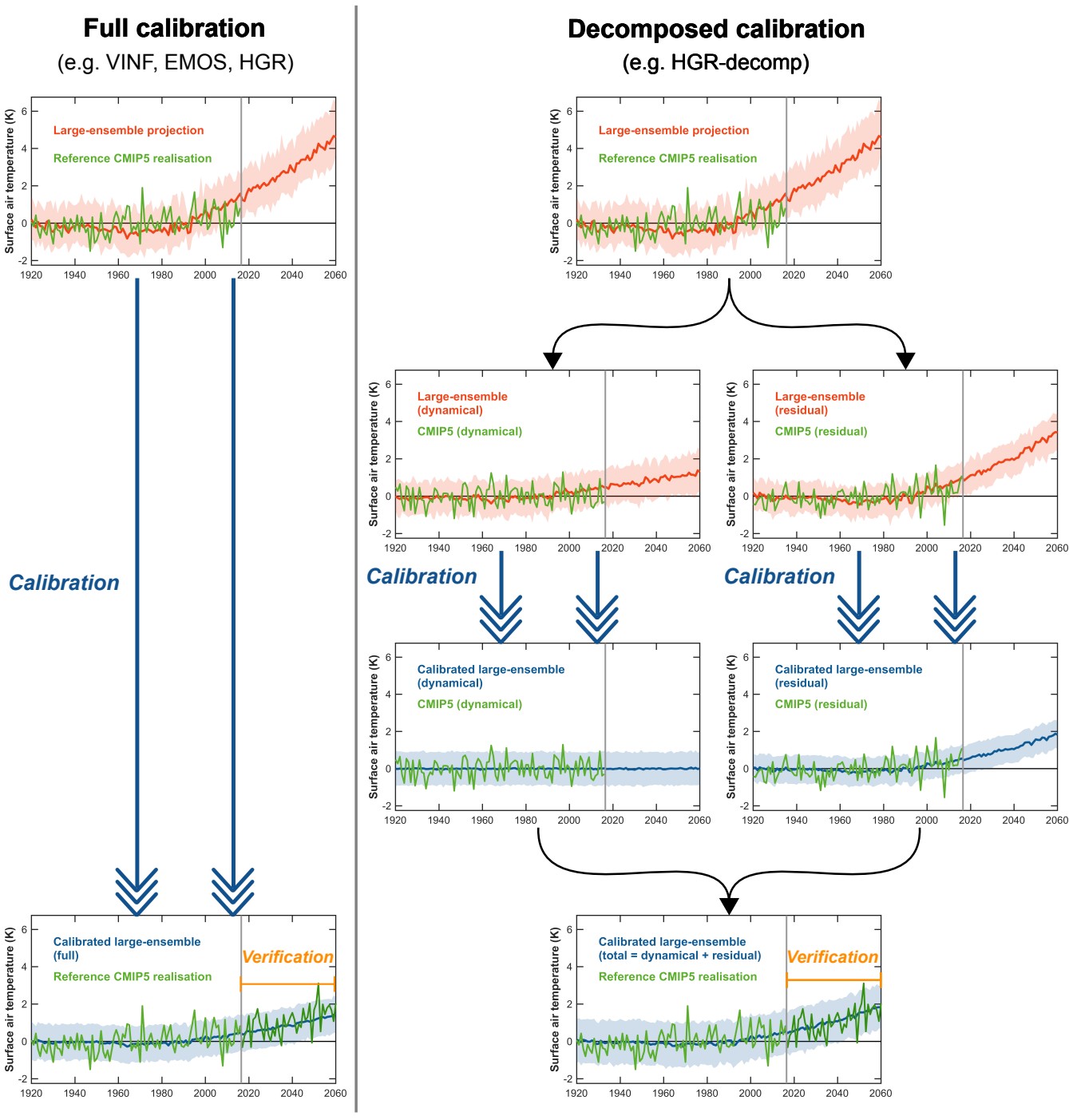

**Figure 3.** An example showing the the steps of the full calibration methods (left hand column) and calibration of the dynamically decomposed variables (right column). This example is the calibration of the summer (JJA) Central European (CEUR) temperature using the CESM1-LE and one of the CMIP5 models. The shading shows the 5-95% range of the CESM1-LE ensemble. The effectiveness of the calibration is assessed by verifying over data from the period 2017-2060, which is withheld during the calibration step.

# CESM1-LE JJA Temperature

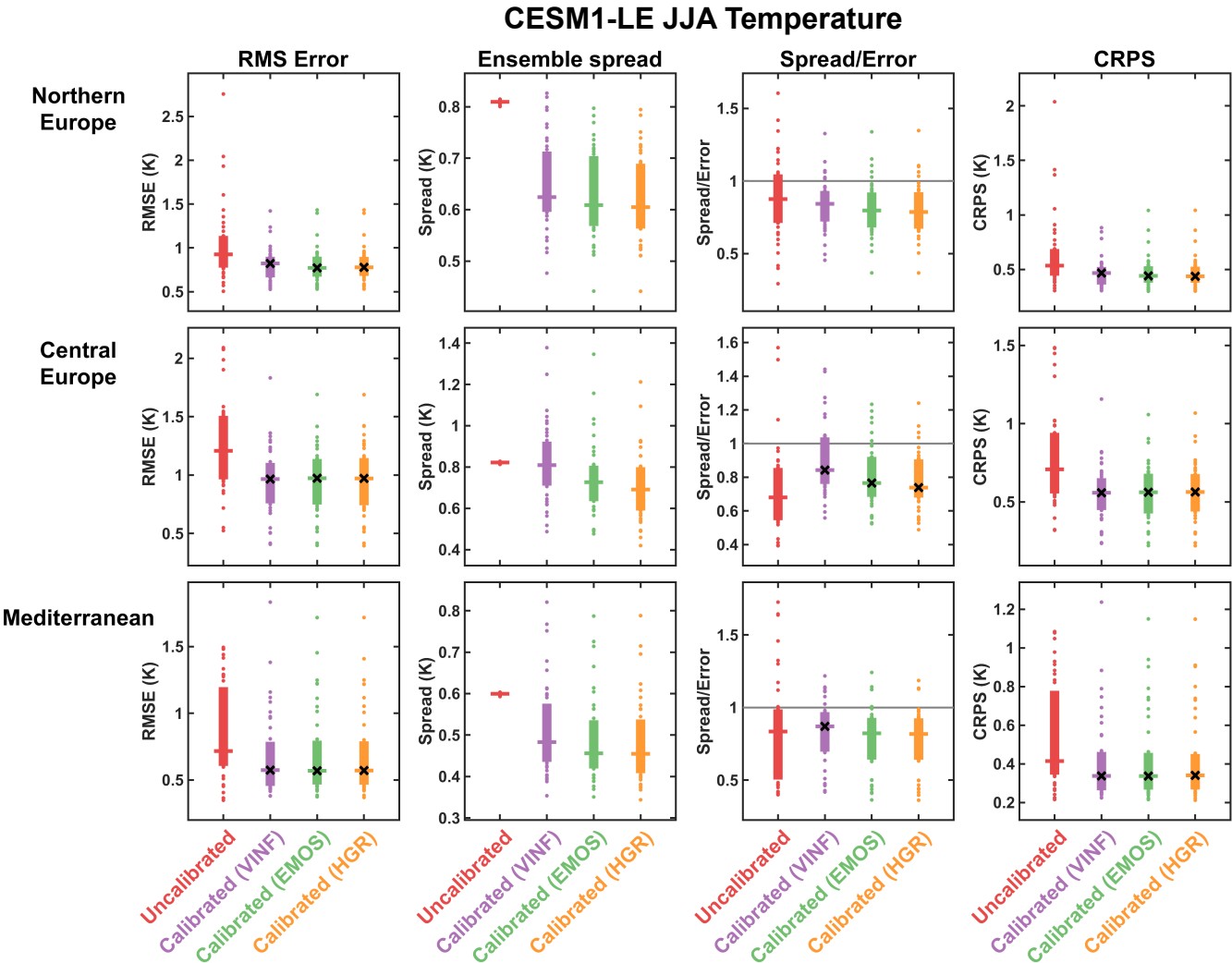

**Figure 4.** Comparison of calibration methods applied to the CESM1-LE summer temperature projections calibrated to the CMIP5 models over the observational period (1920-2016) and verified using the 44 years in the out-of-sample period (2017-2060). The verification statistics for each of the individual CMIP5 models are shown in dots, the interquartile range of this distribution is shown by the solid bars and the median is indicated by the horizontal lines. For the calibrated RMS Error, spread/error and CRPS values, the black crosses indicate where the calibration represents a significant improvement over the uncalibrated (but bias-corrected) ensemble at the 90% significance level. The significance levels were calculated using the non-parametric Mann-Whitney U-test, applied to the distributions of the verification scores from the 39 CMIP5 models.

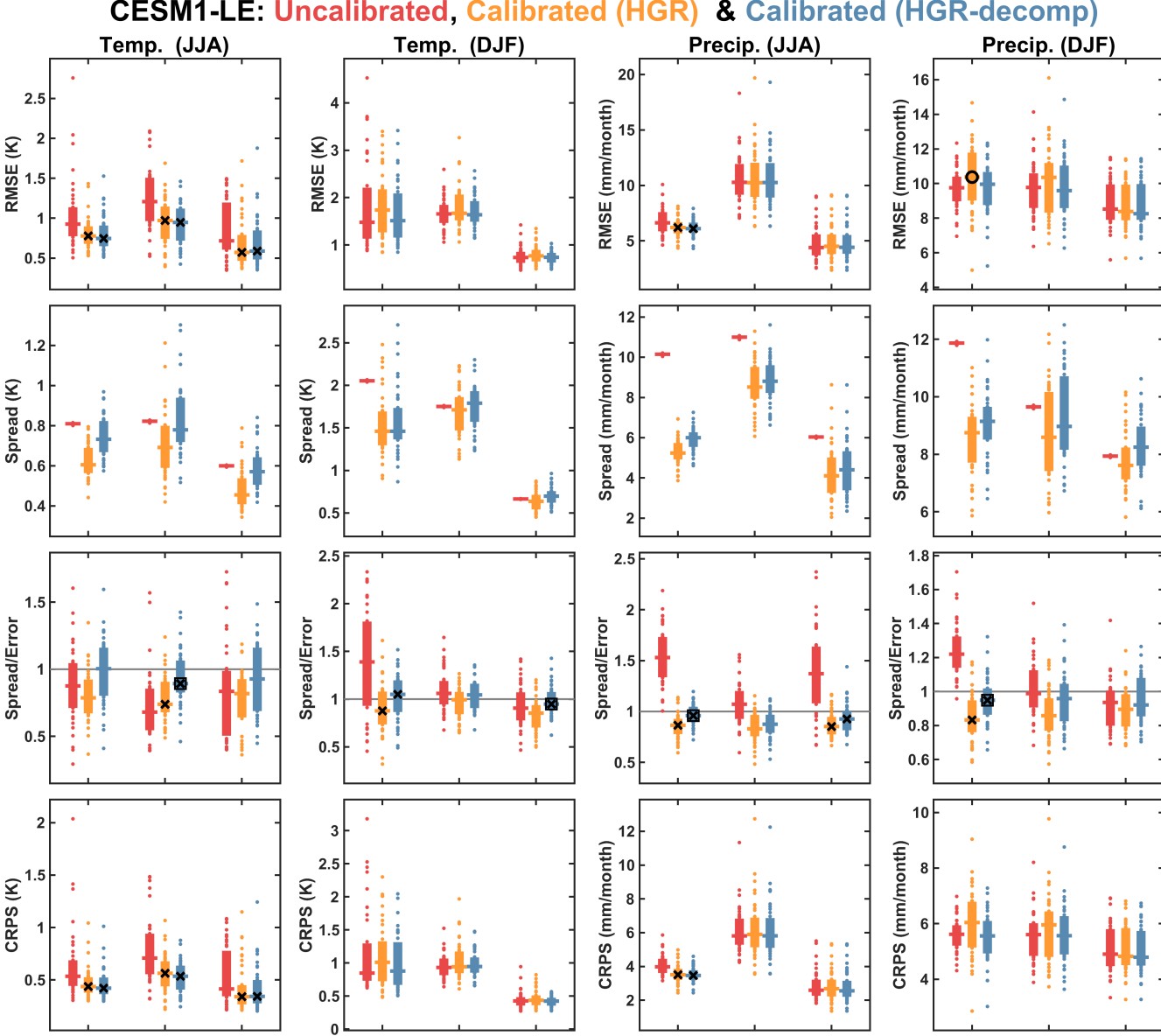

**Figure 5.** Overview of verification of the HGR and HGR-decomp calibration methods compared with the uncalibrated CESM1-LE data in the European regions. Results are shown for all of the verification measures, for both summer and winter seasons and for temperature and precipitation. The verification statistics for each of the individual CMIP5 models are shown in dots, the interquartile range of this distribution is shown by the solid bars and the median is indicated by the horizontal lines. For the calibrated RMS Error, spread/error and CRPS values, the black crosses indicate where the calibration represents a significant improvement over the uncalibrated (but bias-corrected) ensemble at the 90% significance level. Black circles indicate where the calibration is significantly worse than the uncalibrated ensemble (at the 90% level). Black boxes indicate where the HGR-decomp method of calibration is significantly better than the HGR method (at the 90% level). The significance levels were calculated using the non-parametric Mann-Whitney U-test, applied to the distributions of the verification scores from the 39 CMIP5 models.

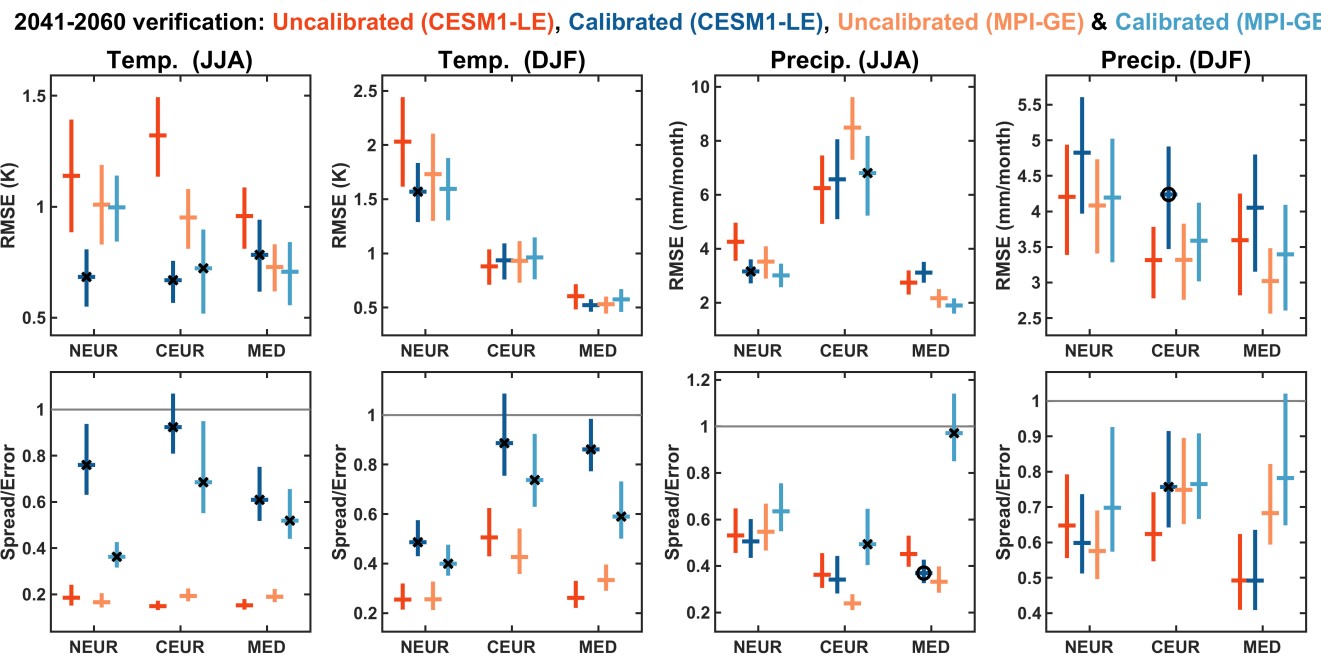

**Figure 6.** Verification of the 2041-2060 mean projections calculated relative to the out-of-sample CMIP5 models for both the CESM1-LE and MPI-GE datasets. The horizontal lines show the mean across all models and the vertical lines show the 90% confidence intervals, calculated by randomly resampling across the CMIP5 models with replacement 1000 times. The black crosses indicate where the calibrated ensemble is significantly better than the equivalent uncalibrated ensemble; the black circles indicate where the calibrated ensemble is significantly worse that the uncalibrated ensemble.

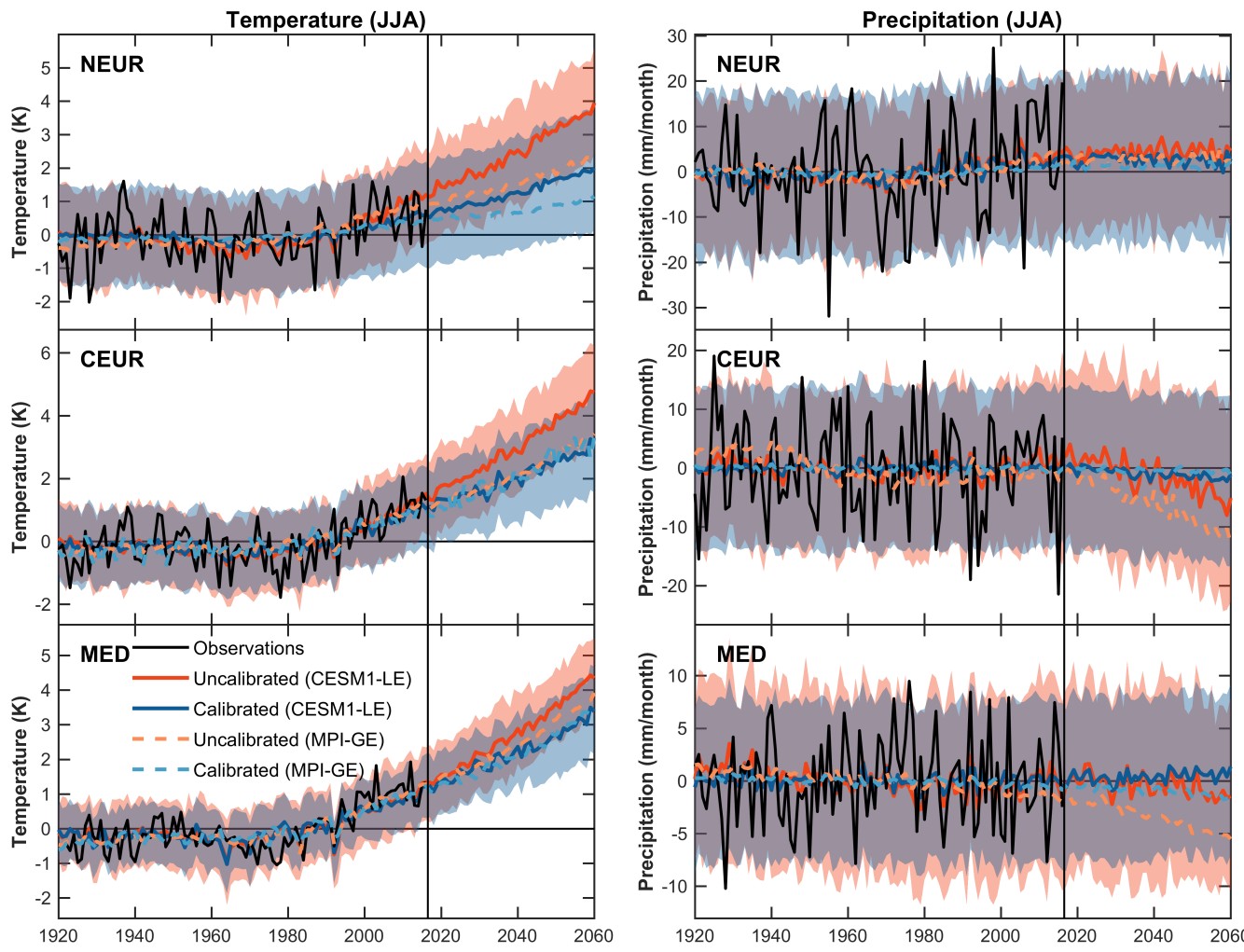

**Figure 7.** Uncalibrated and calibrated (HGR-decomp) CESM1-LE projections, where here the calibrated projections have been calibrated against the observations over the period 1920-2016. The lines show the ensemble medians for the uncalibrated and calibrated ensembles for both the CESM1-LE (solid) and MPI-GE (dashed) datasets. The shading shows the 5-95% range of the CESM1-LE ensemble. Based on the verification out-of-sample tests using the CMIP5 models the calibrated ensemble is expected to be more reliable than the uncalibrated ensemble, particularly for temperatures.

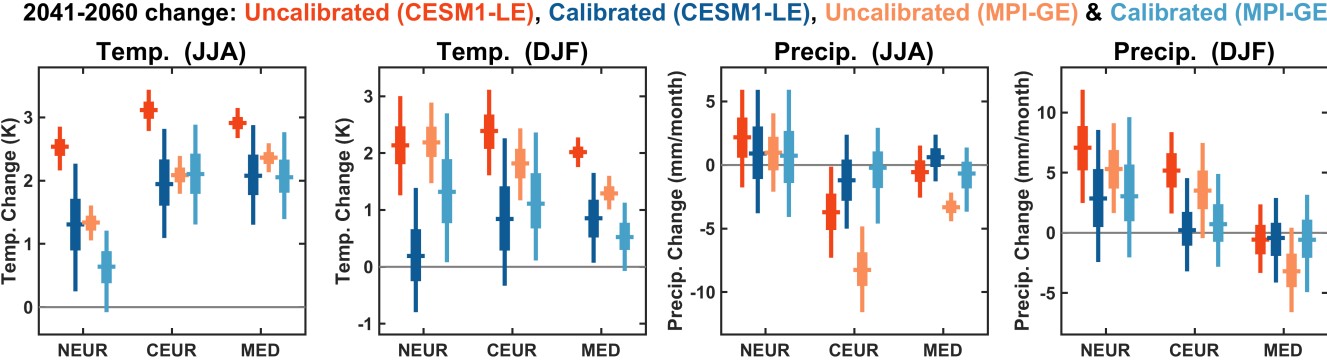

**Figure 8.** 2041-2060 mean calculated relative to 1995-2014 climatology for both CESM1-LE and MPI-GE, calibrated using the HGR-decomp method to the observations over the period 1920-2016. The vertical lines show the 90% range of the ensemble, thick boxes show the interquartile range and horizontal lines show the ensemble median.