# Peer review of "Calibrating large-ensemble European climate projections using observational data"

_Earth System Dynamics, 2020_

## Referee Comment (RC1) · Francisco J. Doblas-Reyes (Referee) · 12 Apr 2020

This is a valuable manuscript that aims to apply ideas common in weather and climate prediction into the post-processing of climate projections, in particular with the use of large ensembles. The authors undertake an ambitious analysis to illustrate the relevance of calibrating the projection ensembles to increase their accuracy and reliability, where reliability is considered from the point of view of the trustworthiness of the probabilities formulated for the ensemble projections. The ideas are solid and clearly laid out, the text is clear, the figures adequate both in number and quality, the study is exhaustive. However, I am concerned by the description of the "out-of-sample with

imperfect model test". The method is explained in page 7 and an example is given in figure 3, but it is hard to understand how the results displayed in figure 4 are obtained. As a result, Figure 4 is a bit hard to interpret. It will benefit from a more detailed caption and better referencing in the main text. Also, the wording and the interpretation of the results can be misleading. For instance, it is hard to accept that the results of the methods lead to improvements when the verification is performed without using observations. It is also a pity that the supplementary information does not include the results equivalent to figure 4 but for precipitation. The HGR-decomp method looks promising. However, it would be really useful if the authors could provide a full illustration of how each component is calibrated before the ensemble is reconstructed, that is, to go beyond what is currently shown in figure 6. This is far from obvious and would help to understand how the method works. Figures 8 and 9 show that the mean projected change is weaker in the calibrated with respect to the uncalibrated large ensembles, particularly for precipitation. This is an important statement, although it comes with a widening of the uncertainty intervals. I wonder how these results compare to other post-processing exercises (like model selection or model weighting) performed with other ensembles in the same areas and period. I consider the manuscript needs major revisions, not that much from the technical or conceptual point of view, but more for the need to clarify some details in the text. Some minor comments follow: - p. 2, l. 24: "applied" appears twice in the sentence. - p. 3, l. 1: "that" appears twice. - p. 3, l. 14-15: This is an interesting idea, although the reader might benefit from more details about how this merging could work and why it's a relevant issue. - p. 4, l. 3-4: To what measure is the regridding affecting the results? Is LENS the ensemble with the coarser resolution? Has the regridding to a different grid been tested? - p. 5, l. 17: Correct "corrlation". Also, the sentence is incomplete. - p. 6, l. 9: Can you say a bit more about the resampling done. For instance, is it performed with or without replacement? - p. 6, l. 13: Use "constant in time". - p. 6, l. 30: Use "to compute". - p. 7, l. 8: Remove "is". - p. 8, l. 18: Correct "signficantly". This mistake appears in other parts of the text. - p. 10, l. 24: How can the reader see the overfit of the HGR method when compared to

the HGR-decomp method? - p. 11, l. 1: This is an example of my main concern with this manuscript. The text mentions an improvement for the projected climate over the period 2041-2060. However, it's hard for me to accept that there is an improvement when no comparison with the observations (which obviously do not exist yet) is made. - p. 11, l. 17: Change "it it calibrated". - p. 11, l. 29-31: It is hard to see any changes in spread in figure 8. - p. 11, l. 32-33: I would not say that the impact of the calibration on the precipitation projections is "fairly modest". - p. 12, l. 6: Correct "preciptation". - p. 13, l. 20-27: This argument seems a bit hard to follow to me. How can we determine if a third calibrated ensemble outperforms or not the former two in terms of future projections? - The figure 4 caption mentions a 44-year verification period starting in 1917, which seems wrong. Also, in the caption the sentence "For the calibrated RMS Error, spread/error and CRPS values, the black crosses indicate where the calibration represents a significant improvement over the uncalibrated (but bias-corrected) ensemble at the 90% significance level" misses to explain what is actually tested: the median of the distribution of calibrated scores, all the scores in a single sample or anything else. Finally, what does the range of values for the uncalibrated ensemble represent? If they haven't been calibrated, do they represent the scores against the CMIP5 single models?

---

## Referee Comment (RC2) · Anonymous Referee #2 · 18 Apr 2020

Review of "Calibrating large-ensemble European climate projections using observational data" by O'Reilly, Befort and Weisheimer, submitted to Earth System Dynamics.

This is an ambitious and novel study aimed at improving climate projections using calibration techniques developed for initialized seasonal prediction. The approaches are tested on two single-model Large Ensembles (LE) using out-of-sample verification methods based on CMIP5 models. The analysis focuses on temperature and precipitation over Europe and takes into account seasonality. Another novel aspect is the application of the calibration method on the dynamical and residual thermodynamic components separately using the technique of "dynamical adjustment". This yields an

improvement in the accuracy of projections of temperature but not precipitation. The study is comprehensive and the methods are scientifically sound. The paper is generally well written, although some clarification is needed in places. I have a number of comments and suggestions as detailed below, but they are mostly minor in scope.

1) P2 L24: remove "was applied to" 2) P2 L32: Perhaps reference Deser et al. (2020) which provides a broader view of the utility of Large Ensembles with multiple models, and includes a more comprehensive listing of LE experiments to date.

Deser, C., F. Lehner, K. B. Rodgers, T. Ault, T. L. Delworth, P. N. DiNezio, A. Fiore, C. Frankignoul, J. C. Fyfe, D. E. Horton, J. E. Kay, R. Knutti, N. S. Lovenduski, J. Marotzke, K. A. McKinnon, S. Minobe, J. Randerson, J. A. Screen, I. R. Simpson and M. Ting, 2020: Insights from earth system model initial-condition large ensembles and future prospects. Nat. Clim. Change, doi: 10.1038/s41558-020-0731-2.

3) P3 L23: Suggest using "CESM1-LE" in place of "LENS" throughout for parallel construction with "MPI-GE". 4) P3 L33: Please do some sensitivity tests on the choice of SLP dataset. I know that HadSLP2 generally has lower amplitude variability (and maybe trends) than 20CR or ERA20C. 5) P4 L19: "lies" should be "lie" 6) P4 L20: "is further" should be "are further" 7) P5 L17: "correlation" is mis-spelled and there is some missing text after "ensemble and, " 8) P6 L4: "time" should be plural 9) P6 L24: Add "Guo et al., 2019" to your list of references (this was an application to precipitation)

Guo, R., C. Deser, L. Terray and F. Lehner, 2019: Human influence on winter precipitation trends (1921-2015) over North America and Eurasia revealed by dynamical adjustment. Geophys. Res. Lett., 46, doi: 10.1029/2018GL081316.

10) P7 L8: "is clearly has a" is not grammatical 11) P8 L19: This sentence is confusing because it sounds like you are only testing the methods on the MPI-GE, but that is not the case. I suggest first discussing the LENS results and then moving on to the MPI results. 12) P9 L2: is the lack of improvement in winter because the characteristics of the variability are not distinguishable between LENS and CMIP5? 13) P9 L 3: "are"

[Figure]

should be "is" 14) P9 L5: "larger than is appropriate": please explain what you mean. Does this imply that LENS has more variability than the other CMIP5 models, or a larger forced signal? Relatedly, it would be very nice to see some discussion of the relevance of the so-called "signal-to-noise paradox" in the seasonal-to-interannual prediction literature for climate change projections. 15) P9 L10: "in to" should be "is to" 16) P9 L18: Change "covarying signal in the reference/observational index" to "covarying signal between the reference and observational indices" for clarity (unless I misunderstand your approach). 17) P9 L21: "with a circulation driven signal": do you want to specify whether this can be an "internal" circulation driven signal, or forced, or both? 18) P10 L20: "separately" is mis-spelled 19) P10 L21: "in the ensemble with a signal": please clarify your intended meaning; the language is confusing. 20) P11 L1: "of temperature.": I would add "in both seasons and models, but especially summer". 21) P11 L27: "from the all of" ? 22) P11 L30-31: Can you provide a physical explanation for why the calibration method acts to increase the uncertainty in future projections? Does it have to do with differences between the level of variability between observations and the model? 23) P12 L9: Is the reduced drying mainly dynamical or thermodynamic in origin? 24) P12 L15 "far more consistent . . .": I think this is an overstatement. 25) P12 L22-30: How do your results relate, if at all, to the trend biases in LENS compared to a synthetic observational Large Ensemble documented in McKinnon and Deser (2018)?

McKinnon, K. A and C. Deser, 2018: Internal variability and regional climate trends in an Observational Large Ensemble. J. Climate, 31, 6783–6802, doi: 10.1175/JCLI-D-17-0901.1.

26) P13 L31: suggest adding "in the calibrated ensembles" after "generally smaller" 27) P14 L7: "For precipitation, where there is no clear signal over the reference period in the observations": I am not sure what your evidence is. Guo et al. (2019) found a nice correspondence with dynamically-adjusted precipitation trends from observations and the ensemble-means of LENS and CMIP5 models. 28) P14 L12: add "relative to the internal variability" after "weaker" (i.e., the forced signal doesn't weaken on smaller

scales, just the signal-to-noise weakens). 29) P14 L21: "is kept" should be "are kept" 30) P14 L27: Cite Yeager et al. (2018) for the LENS DPLE.

Yeager, S. G., G. Danabasoglu, N. Rosenbloom, W. Strand, S. Bates, G. Meehl, A. Karspeck, K. Lindsay, M. C. Long, H. Teng, and N. S. Lovenduski, 2018: Predicting near-term changes in the Earth System: A large ensemble of initialized decadal prediction simulations using the Community Earth System Model, Bull. Amer. Meteor. Soc., in press, doi: 10.1175/BAMS-D-17-0098.1.

31) P14 L28: "merged calibrated climate predictions": insert "set of" before "climate predictions"? 32) Caption to Fig. 3: add "summer" before "temperature" 33) Title to Fig. 4: It is confusing. Suggest re-wording as: "LENS JJA Temperature" (analogous comment applies to Fig. S1). 34) Title to Fig. 5: omit the dash after "LENS" for clarity 35) Caption to Fig. 5: 2nd sentence: change "Shown" to "Results are shown ...". Also, the sentence describing what the black boxes mean is confusing. I would shorten to: "Black boxes indicate where the HGR-decomp method of calibration is significantly better than the HGR method (at the 90% level)." 36) Caption to Fig. 7, line 3: change "has a" to "is". In the next line, change "worse that" to "worse than". 37) Caption to Fig. 8: Please state what the various colors and linestyles mean, and what the shading means. Don't rely on the legend. Indeed, the colors/linestyles in the legend seems to be at odds with that shown in Fig. 7, which had all blue for LENS and all red for MPI. Please make them consistent for clarity. 38) Fig. 9: Same comment as above: please use a consistent color scheme as in Fig. 7 (or change Fig. 7 to be consistent with Fig. 9). 39) Caption to Fig. 9: Please state the method of calibration in the caption. Is it HGR-decomp?

---

## Referee Comment (RC3) · Anonymous Referee #3 · 19 Apr 2020

Summary:

This study takes methods of forecast calibration normally used for initialized seasonal forecasts and applies them to half-century scale regional climate projections. Three similar recalibration methods are applied to two single model large ensembles of RCP 8.5 simulations. The recalibration methods are tested using an imperfect model approach where CMIP5 models are used in place of observations to allow out-of-sample evaluation of the skill of the recalibrated projections. The imperfect model testing indicates that recalibration generally produces more reliable projections for future climate in Europe, and only rarely produces significantly less reliable projections. Results are

qualitatively similar for both large ensembles. An important aspect of this study is the separate recalibration of dynamically decomposed components of the forecasts, which tends to produce more reliable projections that recalibrating the complete forecasts.

I congratulate the authors on presenting a fascinating idea. The manuscript is generally very clear, and I have little criticism of the imperfect model validation methodology which is very thorough. The idea proposed is can uninitialized mid-term climate projections be recalibrated to be more useful for adaptation and impact assessment using techniques from seasonal/decadal forecasting? The answer is almost certainly yes, as this study demonstrates, but with some important caveats that warrant further discussion without detracting from the novelty and potential utility of the idea.

The main concern is conceptual. The three recalibration methods tested are very similar, effectively differing only in their treatment of the ensemble spread. They were conceived for application to seasonal forecasts where uncertainty in the forcing and the thermodynamic response to forcing (i.e., climate change) are negligible. On decadal time scales this assumption may still be a reasonable approximation, but on longer time scales this is not the case, as is clearly visible in Figure 3 by the divergence between the CESM ensemble and the CMIP5 model. The recalibration methods used were not intended to correct for differences in forcing or response to forcing. Therefore, unless the difference over time is approximately constant (which it isn't), or can be corrected by a linear scaling of the signal (Figure 3 suggests not), then the recalibration methods tested are likely to be inadequate to the task. I do not doubt the performance improvements shown in the results, bias correction, signal scaling and correcting the ensemble spread will all improve the imperfect model predictions, but I doubt whether the projections are truly reliable.

This makes the dynamical decomposition aspect of the paper all the more interesting and important. The idea appears to be to decompose the forecasts into forced and unforced components, then recalibrate each component separately using the same recalibration method. This makes a lot of sense and goes a long way to addressing
my concerns above (it is still questionable whether the recalibrations employed are suitable for the forced component, however this is pardonable given the novelty of the approach). In my view, the decomposition step is critical to making the whole approach credible and needs to be introduced and motivated in the introduction, some further details of the both the decomposition itself and how the components are recombined (Figure 6) included in methodology, and possibly some additional reflection in the conclusions.

Specific points:

Page 6, Lines 5-6: Arguably, EMOS is the most general of the three methods. VINF is optimal in mean square error, making it equivalent to EMOS with c=0 when EMOS is optimized on the log score rather than CRPS. Similarly, HGR is equivalent to EMOS with d=0, on the log score.

Page 6, Lines 7-10: Was a block sampling strategy used to account for trends and periodic features such as ENSO? If not this would represent a great deal of work to repeat, so I do not insist it is done, but more details would be helpful.

Page 6, Line 30: computed -> compute

Page 7, Line 8: the raw ensemble is clearly has -> the raw ensemble clearly has

Page 7, Lines 7-9: In apparent contradiction to the text, there is no visible positive bias in the upper panel of Figure 2, and the reference never lies outside of the ensemble.

Page 7, Lines 27-29: It would be useful to have some of these results available in the supplementary material. It seems likely that there will be systematic differences depending on the calibration period, given the relative lack of signal in most models until around 1990, the inability of most CMIP5 models to reproduce the so-called hiatus period, and the fact that the forcing after 2005 will differ from the observations. Longer calibration periods will down-weight the information contained in these key periods.

Page 11, Lines 15-17: Given my primary concern above, and my comment on Page

7, it would also be useful to have some of these results available in supplementary material, and a little more discussion given.
* * *

---

## Referee Comment (RC4) · Anonymous Referee #4 · 30 Apr 2020

This paper presents a novel study which attempts to create better projections by calibrating large ensembles over a calibration period where we have both observations and large ensemble simulations. This study investigates three methods of calibration and finds that while all methods perform well, no method performs substantially better than the others. They then show improvement by using a dynamical decomposition method.

They find that the calibration works much better for temperature than precipitation, and attribute this to the lack of clear forced change in the calibration period for precipitation. For temperature they find improvement for both large ensembles over Europe by using

this calibration method and find that it reduces warming as compared to the calibrated ensemble. I recommend publication with a few minor points to be addressed.

Minor points: Page 3 line 2 should be 'ensembles' Page 3 lines 27/28 MPI-GE is initialized from different years of a long pre-industrial control run, not in the same way as LENS Page 4 line 22 should be 'projections' Section 2.3.1 Are you results sensitive to the choice of reference period? For the dynamical decomposition can you explain why and how you use SLP? Pg 7 lines 7/8. Please explain what you mean by "The raw ensemble is clearly has a positive bias" Section 3.3 The explanation at the beginning of the section should be in Section 2.4

Additional studies that may be of interested: only cite if you feel appropriate. https://www.earth-syst-dynam-discuss.net/esd-2019-69/ https://journals.ametsoc.org/doi/full/10.1175/JCLI-D-16-0905.1 Deser, C., F. Lehner, K. B. Rodgers, T. Ault, T. L. Delworth, P. N. DiNezio, A. Fiore, C. Frankignoul, J. C. Fyfe, D. E. Horton, J. E. Kay, R. Knutti, N. S. Lovenduski, J. Marotzke, K. A. McKinnon, S. Minobe, J. Randerson, J. A. Screen, I. R. Simpson and M. Ting, 2020: Insights from earth system model initial-condition large ensembles and future prospects. Nat. Clim. Change, doi: 10.1038/s41558-020-0731-2. [SharedIt Link]

---

## Author Comment (AC1) · 23 Jun 2020

**Reply to RC1 (comments in blue, reply in black)**

*General/major comments*

*This is a valuable manuscript that aims to apply ideas common in weather and climate prediction into the post-processing of climate projections, in particular with the use of large ensembles. The authors undertake an ambitious analysis to illustrate the relevance of calibrating the projection ensembles to increase their accuracy and reliability, where reliability is considered from the point of view of the trustworthiness of the probabilities formulated for the ensemble projections. The ideas are solid and clearly laid out, the text is clear, the figures adequate both in number and quality, the study is exhaustive. However, I am concerned by the description of the "out-of-sample with imperfect model test". The method is explained in page 7 and an example is given in figure 3, but it is hard to understand how the results displayed in figure 4 are obtained. As a result, Figure 4 is a bit hard to interpret. It will benefit from a more detailed caption and better referencing in the main text. Also, the wording and the interpretation of the results can be misleading. For instance, it is hard to accept that the results of the methods lead to improvements when the verification is performed without using observations. It is also a pity that the supplementary information does not include the results equivalent to figure 4 but for precipitation.*

We agree with the reviewer that it is important to clarify the description of the imperfect model testing. This is central to this study, so we will include an expanded description, including a schematic illustration of the process involved to arrive at the verification statistics presented in the paper. This will result in a clearer presentation of Figure 4 and the related plots. In addition, we will add the equivalent plot for precipitation to the Supplementary Information, as this may be of interest to some readers, as the reviewer rightly highlights.

Regarding the logical step between demonstrating the efficacy of the calibration in the imperfect model tests and extrapolating this when applying to the observations. We of course cannot verify this simply, but one method that might be useful would be to include some analysis of where the parameters of the calibrated observations fits with respect to the CMIP perfect model tests. We will calculate this and include the results in the supplementary material and a discussion detailing this in the revised manuscript.

*The HGR-decomp method looks promising. However, it would be really useful if the authors could provide a full illustration of how each component is calibrated before the ensemble is reconstructed, that is, to go beyond what is currently shown in figure 6. This is far from obvious and would help to understand how the method works.*

We agree, this is a very good suggestion. We will add a schematic to fully illustrate the processes involved, particularly as the methods become more convoluted as the paper goes on. Further discussion will also be added to describe the methodologies in a clearer and more practical manner.

*Figures 8 and 9 show that the mean projected change is weaker in the calibrated with respect to the uncalibrated large ensembles, particularly for precipitation. This is an important statement, although it comes with a widening of the uncertainty intervals. I wonder how these*

*results compare to other post-processing exercises (like model selection or model weighting) performed with other ensembles in the same areas and period. I consider the manuscript needs major revisions, not that much from the technical or conceptual point of view, but more for the need to clarify some details in the text.*

Yes, we agree that the reviewer that the paper would benefit from some discussion of these aspects. We will add discussion and some specifics comparisons with the results for European projections of some other multi-model methods to the revised manuscript (some of these are part of a paper that we are co-authors on and is currently in revision for publication in Journal of Climate).

*Minor comments*

*- p. 2, l. 24: "applied" appears twice in the sentence.*

Yes, this will be corrected.

*- p. 3, l. 1: "that" appears twice.*

Yes, this will be corrected.

*- p. 3, l. 14-15: This is an interesting idea, although the reader might benefit from more details about how this merging could work and why it's a relevant issue.*

Agreed. We will add further details to this idea in the revised manuscript.

*- p. 4, l. 3-4: To what measure is the regridding affecting the results? Is LENS the ensemble with the coarser resolution? Has the regridding to a different grid been tested?*

We have tested this on a small subset of the results and the regridding only marginally affects the results. The LENS ensemble (performed at 1x1 degree resolution in the atmosphere) is generally comparable or higher atmospheric resolution than the CMIP5 models, with 30 vertical levels. The MPI-GE is performed at a relatively low T63 spectral resolution (equivalent to around 2-degree horizontal resolution), with 40 vertical levels. This information will be added to the revised manuscript.

*- p. 5, l. 17: Correct "corrlation". Also, the sentence is incomplete.*

Thanks for spotting this – it was a mistake and will be corrected.

*- p. 6, l. 9: Can you say a bit more about the resampling done. For instance, is it performed with or without replacement?*

The resampling was performed with replacement – this is a relevant detail and will be added to the revised manuscript.

*- p. 6, l. 13: Use "constant in time".*

Agreed, will change in the revised manuscript.

*- p. 6, l. 30: Use "to compute".*

Agreed, will change in the revised manuscript.

*- p. 7, l. 8: Remove "is".*

Agreed, will change in the revised manuscript.

*- p. 8, l. 18: Correct "signficantly". This mistake appears in other parts of the text.*

Agreed, will change in the revised manuscript and check for other occurrences of this mistake.

*- p.10, l. 24: How can the reader see the overfit of the HGR method when compared to the HGR-decomp method?*

Here we were interpreting the relatively low spread in the HGR compared with the HGR-decomp as being due to an overfitting to the reference timeseries – resulting in a consistently lower Spread/Error ratio in the HGR. This interpretation and the justification for it will be added to the revised manuscript.

*- p. 11, l. 1: This is an example of my main concern with this manuscript. The text mentions an improvement for the projected climate over the period 2041-2060. However, it's hard for me to accept that there is an improvement when no comparison with the observations (which obviously do not exist yet) is made.*

As the reviewer suggests, we of course cannot verify this simply, but one method that might be useful would be to include some analysis of where the parameters of the calibrated observations fits with respect to the CMIP perfect model tests. We will calculate this and include the results in the supplementary material and a discussion detailing this in the revised manuscript. In addition, we will edit the text to state more cautiously that the results suggest that this process may result in improved projections but that there are some important caveats.

*- p. 11, l. 17: Change "it it calibrated".*

Agreed, will change in the revised manuscript.

*- p. 11, l. 29-31: It is hard to see any changes in spread in figure 8.*

Agreed, will change in the revised manuscript.

*- p. 11, l. 32-33: I would not say that the impact of the calibration on
the precipitation projections is "fairly modest".*

Agreed, that is not a good description. We will amend in the revised manuscript.

*- p. 12, l. 6: Correct "preciptation".*

*- p. 13, l. 20-27: This argument seems a bit hard to follow to me. How can we determine
if a third calibrated ensemble outperforms or not the former two in terms of future
projections?*

Agreed, will change in the revised manuscript.

*- The figure 4 caption mentions a 44-year verification period starting in 1917,
which seems wrong. Also, in the caption the sentence "For the calibrated RMS Error,
spread/error and CRPS values, the black crosses indicate where the calibration represents
a significant improvement over the uncalibrated (but bias-corrected) ensemble
at the 90% significance level" misses to explain what is actually tested: the median
of the distribution of calibrated scores, all the scores in a single sample or anything
else. Finally, what does the range of values for the uncalibrated ensemble represent?
If they haven't been calibrated, do they represent the scores against the CMIP5 single
models?*

Yes, the year here is a typo and will be changed in the revised manuscript. The significance
testing was performed on the distribution of the verification scores and was tested using the
Mann-Whitney U-test. Further details will be added to the revised manuscript.

The uncalibrated ensemble has only been bias corrected over the reference period (so is not
strictly uncalibrated) but this needs to be stated more clearly and will be corrected in the
revised manuscript.

We thank the reviewer for their insightful and helpful comments that we hope will help to
improve the paper.

---

## Author Comment (AC2) · 23 Jun 2020

**Reply to RC2 (comments in blue, reply in black)**

*General comments*

*This is an ambitious and novel study aimed at improving climate projections using calibration techniques developed for initialized seasonal prediction. The approaches are tested on two single-model Large Ensembles (LE) using out-of-sample verification methods based on CMIP5 models. The analysis focuses on temperature and precipitation over Europe and takes into account seasonality. Another novel aspect is the application of the calibration method on the dynamical and residual thermodynamic components separately using the technique of "dynamical adjustment". This yields an improvement in the accuracy of projections of temperature but not precipitation. The study is comprehensive and the methods are scientifically sound. The paper is generally well written, although some clarification is needed in places. I have a number of comments and suggestions as detailed below, but they are mostly minor in scope.*

We thank the reviewer for their positive comments and feedback. We agree that there are some aspects of the paper that require clarification and would benefit from further discussion in the revised manuscript – further details follow the specific points below.

*Specific comments*

*1) P2 L24: remove "was applied to"*

Agreed, will change in the revised manuscript.

*2) P2 L32: Perhaps reference Deser et al. (2020) which provides a broader view of the utility of Large Ensembles with multiple models, and includes a more comprehensive listing of LE experiments to date. Deser, C., F. Lehner, K. B. Rodgers, T. Ault, T. L. Delworth, P. N. DiNezio, A. Fiore, C. Frankignoul, J. C. Fyfe, D. E. Horton, J. E. Kay, R. Knutti, N. S. Lovenduski, J. Marotzke, K. A. McKinnon, S. Minobe, J. Randerson, J. A. Screen, I. R. Simpson and M. Ting, 2020: Insights from earth system model initial-condition large ensembles and future prospects. Nat. Clim. Change, doi: 10.1038/s41558-020-0731-2.*

Yes, this is an important reference. I think (or hope) that this was published after submission but is clearly a very relevant and useful reference and will be included in the revised manuscript.

*3) P3 L23: Suggest using "CESM1-LE" in place of "LENS" throughout for parallel construction with "MPI-GE".*

Agreed, this is neater and we will make this change in the revised manuscript.

*4) P3 L33: Please do some sensitivity tests on the choice of SLP dataset. I know that HadSLP2 generally has lower amplitude variability (and maybe trends) than 20CR or ERA20C.*

We will perform some sensitivity tests and include details in the revised manuscript.

*5) P4 L19: "lies" should be "lie"*

Yes, this will be corrected.

*6) P4 L20: "is further" should be "are further"*

Yes, this will be corrected.

*7) P5 L17: "correlation" is mis-spelled and there is some missing text after "ensemble and, "*

Yes, the spelling and text in this passage will be corrected in the revised manuscript.

*8) P6 L4: "time" should be plural*

Agreed, will change in the revised manuscript.

*9) P6 L24: Add "Guo et al., 2019" to your list of references (this was an application to precipitation) Guo, R., C. Deser, L. Terray and F. Lehner, 2019: Human influence on winter precipitation trends (1921-2015) over North America and Eurasia revealed by dynamical adjustment. Geophys. Res. Lett., 46, doi: 10.1029/2018GL081316.*

Good point - this was an oversight on our part and is a very relevant paper. A reference to this will be included in the revised manuscript.

*10) P7 L8: "is clearly has a" is not grammatical*

Agreed, will change in the revised manuscript.

*11) P8 L19: This sentence is confusing because it sounds like you are only testing the methods on the MPI-GE, but that is not the case. I suggest first discussing the LENS results and then moving on to the MPI results.*

This is a good suggestion – we will edit this passage accordingly in the revised manuscript.

*12) P9 L2: is the lack of improvement in winter because the characteristics of the variability are not distinguishable between LENS and CMIP5?*

That is the case in the MED region but more generally it might be because there is generally less forced change, so that the internal variability component is more important to calibrate. In NEUR for example, this lead to a clear improvement in the reliability despite no change in the RMSE (e.g. Figure 5). The text will be edited to clarify this point in the revised manuscript.

*13) P9 L 3: "are" should be "is"*

Agreed, will change in the revised manuscript.

*14) P9 L5: "larger than is appropriate": please explain what you mean. Does this imply that LENS has more variability than the other CMIP5 models, or a larger forced signal? Relatedly, it would be very nice to see some discussion of the relevance of the so-called "signal-to-noise paradox" in the seasonal-to-interannual prediction literature for climate change projections.*

As highlighted by the reviewer, it is not correct to say that the spread is "larger than appropriate" because it just means it is larger than the other CMIP5 models. This will be amended in the revised manuscript. There may certainly be some aspects of the "signal-to-noise paradox" which are relevant and have implications for climate projectiosn and we will try to highlight this in the discussion in the revised manuscript.

*15) P9 L10: "in to" should be "is to"*

Agreed, will change in the revised manuscript.

*16) P9 L18: Change "covarying signal in the reference/observational index" to "covarying signal between the reference and observational indices" for clarity (unless I misunderstand your approach).*

Yes, this should certainly have been clearer. The meaning here is to highlight the covarying signal between the reference and the ensemble mean which is being calibrated. This will be clarified in the revised manuscript.

*17) P9 L21: "with a circulation driven signal": do you want to specify whether this can be an "internal" circulation driven signal, or forced, or both?*

Agreed, this could be both and is important to make that clear here – will amend in the revised manuscript.

*18) P10 L20: "separately" is mis-spelled*

Yes, this will be corrected.

*19) P10 L21: "in the ensemble with a signal": please clarify your intended meaning; the language is confusing.*

Agreed, will change in the revised manuscript.

*20) P11 L1: "of temperature.": I would add "in both seasons and models, but especially summer".*

Yes, that is a good suggestion, thanks. Will amend this in the revised manuscript.

*21) P11 L27: "from the all of" ?*

This is will be corrected in the revised manuscript.

*22) P11 L30-31: Can you provide a physical explanation for why the calibration method acts to increase the uncertainty in future projections? Does it have to do with differences between the level of variability between observations and the model?*

Yes, it seems to be largely due to the differences in the levels of variability. We will look into this further though and try to elaborate on this in the revised manscript.

*23) P12 L9: Is the reduced drying mainly dynamical or thermodynamic in origin?*

It seems to be mainly dynamical. In the models there is a stronger dynamical signal over the reference period which doesn't seem to be there in the observations and this is reduced by the calibration. This detail will be added to the revised manuscript.

*24) P12 L15 "far more consistent . . .": I think this is an overstatement.*

Agreed, the "far" is probably not justified here. This will be amended in the revised manuscript.

*25) P12 L22-30: How do your results relate, if at all, to the trend biases in LENS compared to a synthetic observational Large Ensemble documented in McKinnon and Deser (2018)? McKinnon, K. A and C. Deser, 2018: Internal variability and regional climate trends in an Observational Large Ensemble. J. Climate, 31, 6783–6802, doi: 10.1175/JCLI-D-17-0901.1.*

Very interesting question. Thinking about it, in some sense the calibration is "trying" to account of some of these biases but how is related to the trends is not obvious. Nonetheless, this is an important study and discussion of this will be added to the revised manuscript.

*26) P13 L31: suggest adding "in the calibrated ensembles" after "generally smaller"*

Agreed, will change in the revised manuscript.

*27) P14 L7: "For precipitation, where there is no clear signal over the reference period in the observations": I am not sure what your evidence is. Guo et al. (2019) found a nice correspondence with dynamically-adjusted precipitation trends from observations and the ensemble-means of LENS and CMIP5 models.*

Here the statement is just for the seasons and regions specifically analysed in the paper and when comparing the signal to interannual timescale variability (e.g. black lines in Figure 8). It will be clarified that this is not a general statement in the revised manuscript.

*28) P14 L12: add "relative to the internal variability" after "weaker" (i.e., the forced signal doesn't weaken on smaller scales, just the signal-to-noise weakens).*

Agreed, this will be added in the revised manuscript.

*29) P14 L21: "is kept" should be "are kept"*

This is will be corrected in the revised manuscript.

*30) P14 L27: Cite Yeager et al. (2018) for the LENS DPLE. Yeager, S. G., G. Danabasoglu, N. Rosenbloom, W. Strand, S. Bates, G. Meehl, A. Karspeck, K. Lindsay, M. C. Long, H. Teng, and N. S. Lovenduski, 2018: Predicting near-term changes in the Earth System: A large ensemble of initialized decadal prediction simulations using the Community Earth System Model, Bull. Amer. Meteor. Soc., in press, doi: 10.1175/BAMS-D-17-0098.1.*

Good point – this reference will be included in the revised manuscript.

*31) P14 L28: "merged calibrated climate predictions": insert "set of" before "climate predictions"?*

Agreed, this will be added in the revised manuscript.

*32) Caption to Fig. 3: add "summer" before "temperature"*

Agreed, will change in the revised manuscript.

*33) Title to Fig. 4: It is confusing. Suggest re-wording as: "LENS JJA Temperature" (analogous comment applies to Fig. S1).*

Good suggestion – this will be changed here and in Fig S1 in the revised manuscript.

*34) Title to Fig. 5: omit the dash after "LENS" for clarity*

Agreed, will change in the revised manuscript.

*35) Caption to Fig. 5: 2nd sentence: change "Shown" to "Results are shown . . ." .*
*Also, the sentence describing what the black boxes mean is confusing. I would shorten*
*to: "Black boxes indicate where the HGR-decomp method of calibration is significantly*
*better than the HGR method (at the 90% level)."*

Thanks for this suggestion. We will change accordingly in the revised manuscript.

*36) Caption to Fig. 7, line 3: change "has a" to "is". In the next line, change "worse that"*
*to "worse than".*

Agreed, will change in the revised manuscript.

*37) Caption to Fig. 8: Please state what the various colors and linestyles mean, and what the*
*shading means. Don't rely on the legend. Indeed, the colors/linestyles in the legend seems to*
*be at odds with that shown in Fig. 7, which had all blue for LENS and all red for MPI. Please*
*make them consistent for clarity.*

Good point – the inconsistency in colours is quite stupid really and we will change this in the
revised manuscript.

*38) Fig. 9: Same comment as above: please use a consistent color scheme as in Fig. 7 (or*
*change Fig. 7 to be consistent with Fig. 9).*

As above - we will change this in the revised manuscript.

*39) Caption to Fig. 9: Please state the method of calibration in the caption. Is it HGR-*
*decomp?*

Yes, it is HGR-decomp. This will be clarified in the revised manuscript.

We thank the reviewer for their incredibly helpful review. There were lots of insightful and
helpful comments and we are confident these will help to improve the paper.

---

## Author Comment (AC3) · 23 Jun 2020

**Reply to RC3 (comments in blue, reply in black)**

*General comments*

*This study takes methods of forecast calibration normally used for initialized seasonal forecasts and applies them to half-century scale regional climate projections. Three similar recalibration methods are applied to two single model large ensembles of RCP 8.5 simulations. The recalibration methods are tested using an imperfect model approach where CMIP5 models are used in place of observations to allow out-of-sample evaluation of the skill of the recalibrated projections. The imperfect model testing indicates that recalibration generally produces more reliable projections for future climate in Europe, and only rarely produces significantly less reliable projections. Results are qualitatively similar for both large ensembles. An important aspect of this study is the separate recalibration of dynamically decomposed components of the forecasts, which tends to produce more reliable projections that recalibrating the complete forecasts.*

*I congratulate the authors on presenting a fascinating idea. The manuscript is generally very clear, and I have little criticism of the imperfect model validation methodology which is very thorough. The idea proposed is can uninitialized mid-term climate projections be recalibrated to be more useful for adaptation and impact assessment using techniques from seasonal/decadal forecasting? The answer is almost certainly yes, as this study demonstrates, but with some important caveats that warrant further discussion without detracting from the novelty and potential utility of the idea.*

We thank the reviewer for their positive comments and feedback. We agree that there are some important caveats and that these would benefit from further discussion in the revised manuscript – further details follow the specific points below.

*The main concern is conceptual. The three recalibration methods tested are very similar, effectively differing only in their treatment of the ensemble spread. They were conceived for application to seasonal forecasts where uncertainty in the forcing and the thermodynamic response to forcing (i.e., climate change) are negligible. On decadal time scales this assumption may still be a reasonable approximation, but on longer time scales this is not the case, as is clearly visible in Figure 3 by the divergence between the CESM ensemble and the CMIP5 model. The recalibration methods used were not intended to correct for differences in forcing or response to forcing. Therefore, unless the difference over time is approximately constant (which it isn't), or can be corrected by a linear scaling of the signal (Figure 3 suggests not), then the recalibration methods tested are likely to be inadequate to the task. I do not doubt the performance improvements shown in the results, bias correction, signal scaling and correcting the ensemble spread will all improve the imperfect model predictions, but I doubt whether the projections are truly reliable.*

Yes, the reviewer makes a very valid point. As we go to longer lead-times (i.e. further into the future) the errors are expected to get larger, as the error in the scaling will be amplified and the contribution of internal variability reduced. We focused on this mid-century timescale because that is the focus of our current project, however, it is important to assess how the effectiveness of the calibration changes with lead-time. We will calculate the verification over some additional future periods (e.g. 2061-2080) to examine this and include the

discussion of these in the revised manuscript (though the results may end up in the supplementary material as the paper is already fairly lengthy).

*This makes the dynamical decomposition aspect of the paper all the more interesting and important. The idea appears to be to decompose the forecasts into forced and unforced components, then recalibrate each component separately using the same recalibration method. This makes a lot of sense and goes a long way to addressing my concerns above (it is still questionable whether the recalibrations employed are suitable for the forced component, however this is pardonable given the novelty of the approach). In my view, the decomposition step is critical to making the whole approach credible and needs to be introduced and motivated in the introduction, some further details of the both the decomposition itself and how the components are recombined (Figure 6) included in methodology, and possibly some additional reflection in the conclusions.*

The reviewer is right to suggest that the description/presentation of the decomposition method should have been clearer – and this is also reflected in comments by the other reviewers. In the revised manuscript we will include an expanded motivation and description of the method, as well as a schematic showing the specific steps involved in the decomposition and calibration. We agree that this is an important part of our study that was perhaps not illuminated as it might have been in the previous version of the paper.

**Specific points:**

*Page 6, Lines 5-6: Arguably, EMOS is the most general of the three methods. VINF is optimal in mean square error, making it equivalent to EMOS with c=0 when EMOS is optimized on the log score rather than CRPS. Similarly, HGR is equivalent to EMOS with d=0, on the log score.*

Yes, that's a good point that EMOS is the most general. We thank the reviewer for making this point – and for suggesting the other comparisons between the methods. These details will be added to this section of the revised manuscript.

*Page 6, Lines 7-10: Was a block sampling strategy used to account for trends and periodic features such as ENSO? If not this would represent a great deal of work to repeat, so I do not insist it is done, but more details would be helpful.*

Thanks for the suggestion, yes more details would be helpful. The bootstrap resampling was to account for uncertainty in the fit parameters of the calibrations, not to specifically account for periodic features such as ENSO, however, it's likely that the resampling method does implicitly account for some. More details of the method will be added to the revised manuscript.

*Page 6, Line 30: computed -> compute*

Yes, this will be corrected.

*Page 7, Line 8: the raw ensemble is clearly has -> the raw ensemble clearly has*

Yes, this will be corrected.

*Page 7, Lines 7-9: In apparent contradiction to the text, there is no visible positive bias in the upper panel of Figure 2, and the reference never lies outside of the ensemble.*

Agreed - this is a mistake and will be corrected (this comment was in reference to a previous version of this figure that has since been replaced but we should have caught this).

*Page 7, Lines 27-29: It would be useful to have some of these results available in the supplementary material. It seems likely that there will be systematic differences depending on the calibration period, given the relative lack of signal in most models until around 1990, the inability of most CMIP5 models to reproduce the so-called hiatus period, and the fact that the forcing after 2005 will differ from the observations. Longer calibration periods will down-weight the information contained in these key periods.*

Good point - we did do some sensitivity tests and will include some examples of these in the supplementary material when we revise the manuscript.

*Page 11, Lines 15-17: Given my primary concern above, and my comment on Page 7, it would also be useful to have some of these results available in supplementary material, and a little more discussion given.*

Again, this is a fair point and something we will address. The verification statistics over the different period are likely important and we will provide more of these in the supplementary material of the revised manuscript, along with some discussion of these results in the main text.

We thank the reviewer for their insightful and helpful comments that we hope will help to improve the paper.

---

## Author Comment (AC4) · 23 Jun 2020

**Reply to RC4 (comments in blue, reply in black)**

*Summary*

*This paper presents a novel study which attempts to create better projections by calibrating large ensembles over a calibration period where we have both observations and large ensemble simulations. This study investigates three methods of calibration and finds that while all methods perform well, no method performs substantially better than the others. They then show improvement by using a dynamical decomposition method. They find that the calibration works much better for temperature than precipitation, and attribute this to the lack of clear forced change in the calibration period for precipitation. For temperature they find improvement for both large ensembles over Europe by using this calibration method and find that it reduces warming as compared to the calibrated ensemble. I recommend publication with a few minor points to be addressed.*

We thank the reviewer for the positive comments on our study.

*Minor points:*

*Page 3 line 2 should be 'ensembles'*

This will be corrected.

*Page 3 lines 27/28 MPI-GE is initialized from different years of a long pre-industrial control run, not in the same way as LENS*

This is an important distinction and was an oversight on our part. A description to this effect will be added in the revised manuscript.

*Page 4 line 22 should be 'projections'*

This will be corrected.

*Section 2.3.1 Are you results sensitive to the choice of reference period? For the dynamical decomposition can you explain why and how you use SLP?*

No the results are not very sensitive to the reference period. We tested from 30-years up to the full 97-year periods and the verification statistics generally improve with the length of the period, which is why we use the full reference period here. Text describing these tests will be added to the revised manuscript.

The SLP is used to estimate what seasonally anomalies can be attribute to large-scale circulation anomalies (assessed in terms of SLP anomalies). This will be clarified in the revised manuscript, including a schematic illustrating how the dynamical decomposition is applied to produce the calibrated projections.

*Page 7 lines 7/8. Please explain what you mean by "The raw ensemble is clearly has a positive bias"*

This refers to the observations over the reference period – this will be clarified.

*Section 3.3 The explanation at the beginning of the section should be in Section 2.4*

Agreed, the description in section 2.4 will be expanded in the revised manuscript and will also include a schematic to visualise how this is used to produce calibrated projections.

*Additional studies that may be of interested: only cite if you feel appropriate.*
*https://www.earth-syst-dynam-discuss.net/esd-2019-69/*
*https://journals.ametsoc.org/doi/full/10.1175/JCLI-D-16-0905.1*
*Deser, C., F. Lehner, K. B. Rodgers, T. Ault, T. L. Delworth, P. N. DiNezio, A. Fiore, C. Frankignoul, J. C. Fyfe, D. E. Horton, J. E. Kay, R. Knutti, N. S. Lovenduski, J. Marotzke, K. A. McKinnon, S. Minobe, J. Randerson, J. A. Screen, I. R. Simpson and M. Ting, 2020: Insights from earth system model initial-condition large ensembles and future prospects. Nat. Clim.Change, doi: 10.1038/s41558-020-0731-2.*

Agreed, the description in section 2.4 will be expanded in the revised manuscript.

We thank the reviewer for their very useful comments and suggestions.

---

## Referee Report (RR1)

I thank the authors for addressing my earlier comments, particularly for the additional results included in the supplementary material. I have no further comments.